# Competition between kinesin-1 and myosin-V defines *Drosophila* posterior determination

Wen Lu[1], Margot Lakonishok[1], Rong Liu[2], Neil Billington[2], Ashley Rich[3], Michael Glotzer[3], James R Sellers[2], Vladimir I Gelfand[1]*

[1]Department of Cell and Developmental Biology, Feinberg School of Medicine, Northwestern University, Chicago, United States; [2]Cell Biology and Physiology Center, National Heart, Lung and Blood Institute, National Institutes of Health, Bethesda, United States; [3]Department of Molecular Genetics and Cell Biology, University of Chicago, Chicago, United States

**Abstract** Local accumulation of *oskar (osk)* mRNA in the *Drosophila* oocyte determines the posterior pole of the future embryo. Two major cytoskeletal components, microtubules and actin filaments, together with a microtubule motor, kinesin-1, and an actin motor, myosin-V, are essential for *osk* mRNA posterior localization. In this study, we use Staufen, an RNA-binding protein that colocalizes with *osk* mRNA, as a proxy for *osk* mRNA. We demonstrate that posterior localization of *osk*/Staufen is determined by competition between kinesin-1 and myosin-V. While kinesin-1 removes *osk*/Staufen from the cortex along microtubules, myosin-V anchors *osk*/Staufen at the cortex. Myosin-V wins over kinesin-1 at the posterior pole due to low microtubule density at this site, while kinesin-1 wins at anterior and lateral positions because they have high density of cortically-anchored microtubules. As a result, posterior determinants are removed from the anterior and lateral cortex but retained at the posterior pole. Thus, posterior determination of *Drosophila* oocytes is defined by kinesin-myosin competition, whose outcome is primarily determined by cortical microtubule density.

*For correspondence:
vgelfand@northwestern.edu

Competing interests: The authors declare that no competing interests exist.

## Introduction

Kinesin-1, also known as conventional kinesin, is a major microtubule motor that transports many types of cargoes, including mRNA-containing granules towards plus-ends of microtubules (*Verhey and Hammond, 2009*; *Hirokawa et al., 2009*). Kinesin-1 is required for the posterior accumulation of *osk* mRNA in *Drosophila* oocytes that is essential for embryo posterior determination (*Brendza et al., 2000*; *Palacios and St Johnston, 2002*; *Krauss et al., 2009*). Defects in *osk* mRNA localization prevent germ cell formation and abdomen specification (*Lehmann and Nüsslein-Volhard, 1986*). Posterior localization of *osk* mRNA is initially established during mid-oogenesis (late stage 8 to stage 9) and maintained throughout late oogenesis (stage 10B to stage 14). Staufen, an RNA-binding protein that forms ribonucleoprotein particles (RNPs) with *osk* mRNA, is commonly used as a proxy for *oskar* RNA (referred as '*osk*/Staufen' hereafter) (*Brendza et al., 2000*; *Palacios and St Johnston, 2002*; *Erdélyi et al., 1995*; *Shulman et al., 2000*; *Lu et al., 2016*; *Lu et al., 2018*; *St Johnston et al., 1991*).

During mid-oogenesis (stage 7–9), the oocyte forms an anterior-to-posterior gradient of cortical microtubules (*Theurkauf et al., 1992*; *Clark et al., 1994*; *Clark et al., 1997*). These microtubules are anchored at the cortex by their minus ends via the minus-end binding protein Patronin and a microtubule-actin crosslinker, Short stop (Shot) (*Nashchekin et al., 2016*). Because Shot is excluded from the posterior cortex in a Par-1-dependent manner, more microtubule minus ends are anchored

**eLife digest** One of the most fundamental steps of embryonic development is deciding which end of the body should be the head, and which should be the tail. Known as 'axis specification', this process depends on the location of genetic material called mRNAs. In fruit flies, for example, the tail-end of the embryo accumulates an mRNA called *oskar*. If this mRNA is missing, the embryo will not develop an abdomen.

The build-up of *oskar* mRNA happens before the egg is even fertilized and depends on two types of scaffold proteins in the egg cell called microtubules and microfilaments. These scaffolds act like 'train tracks' in the cell and have associated protein motors, which work a bit like trains, carrying cargo as they travel up and down along the scaffolds. For microtubules, one of the motors is a protein called kinesin-1, whereas for microfilaments, the motors are called myosins.

Most microtubules in the egg cell are pointing away from the membrane, while microfilament tracks form a dense network of randomly oriented filaments just underneath the membrane. It was already known that kinesin-1 and a myosin called myosin-V are important for localizing *oskar* mRNA to the posterior of the egg. However, it was not clear why the mRNA only builds up in that area.

To find out, Lu et al. used a probe to track *oskar* mRNA, while genetically manipulating each of the motors so that their ability to transport cargo changed. Modulating the balance of activity between the two motors revealed that kinesin-1 and myosin-V engage in a tug-of-war inside the egg: myosin-V tries to keep *oskar* mRNA underneath the membrane of the cell, while kinesin-1 tries to pull it away from the membrane along microtubules. The winner of this molecular battle depends on the number of microtubule tracks available in the local area of the cell. In most parts of the cell, there are abundant microtubules, so kinesin-1 wins and pulls *oskar* mRNA away from the membrane. But at the posterior end of the cell there are fewer microtubules, so myosin-V wins, allowing *oskar* mRNA to localize in this area. Artificially 'shaving' some microtubules in a local area immediately changed the outcome of this tug-of-war creating a build-up of *oskar* mRNA in the 'shaved' patch.

This is the first time a molecular tug-of-war has been shown in an egg cell, but in other types of cell, such as neurons and pigment cells, myosins compete with kinesins to position other molecular cargoes. Understanding these processes more clearly sheds light not only on embryo development, but also on cell biology in general.

at the anterior and lateral cortex than at the posterior cortex (*Shulman et al., 2000*; *Nashchekin et al., 2016*; *Zimyanin et al., 2008*; *Parton et al., 2011*). This anterior-posterior cortical microtubule gradient is established in stage 7–8 oocytes (*Theurkauf et al., 1992*), just before *osk*/Staufen starts accumulating at the posterior pole. The anterior-posterior gradient of cortical microtubule results in slightly more microtubule plus ends oriented towards the posterior pole (*Nashchekin et al., 2016*; *Zimyanin et al., 2008*; *Parton et al., 2011*; *Khuc Trong et al., 2015*). Kinesin-1 has been proposed to drive *osk* mRNA transport along these weakly biased cortical microtubules, favoring *osk* mRNA movement from the anterior to the posterior pole (*Brendza et al., 2000*; *Palacios and St Johnston, 2002*; *Zimyanin et al., 2008*; *Khuc Trong et al., 2015*; *Nieuwburg et al., 2017*).

In addition to its conventional cargo-transporting activity, kinesin-1 has another essential function in reorganizing microtubules. Using its N-terminal motor domain and a second microtubule-binding site at the C-terminus of kinesin-1 heavy chain (KHC), kinesin-1 slides antiparallel microtubules against each other in many cell types (*Jolly et al., 2010*; *Lu et al., 2013*; *Lu et al., 2015*; *Winding et al., 2016*). Kinesin-driven microtubule sliding together with kinesin-driven cargo transport generates the force that drives fast cytoplasmic streaming of the ooplasm in late-stage *Drosophila* oocytes (*Lu et al., 2016*). Diffusion facilitated by streaming, rather than directed transport along microtubules, is responsible for the localization of posterior determinants during late oogenesis (*Lu et al., 2018*; *Glotzer et al., 1997*; *Forrest and Gavis, 2003*).

Despite the necessity of directed transport and streaming, they are not sufficient for stable *osk*/Staufen accumulation at the posterior pole; an anchorage mechanism is required to counteract both diffusion and shear forces of the streaming ooplasm that can displace *osk*/Staufen from the posterior cap. *osk* mRNA anchorage has been suggested to be actin-dependent. Cortical localization of *osk*/

Staufen is significantly reduced upon F-actin fragmentation (*Cha et al., 2002*). An actin motor, myosin-V (*didum* in *Drosophila*), is involved in *osk* mRNA cortical localization (*Krauss et al., 2009*; *Sinsimer et al., 2013*). However, myosin-V is uniformly localized at the oocyte cortex (*Krauss et al., 2009*); it is not clear why a uniformly distributed cortical anchor attaches *osk*/Staufen only at the posterior pole.

Genetic data suggest that kinesin-1 activity antagonizes the function of myosin-V as a cortical anchor (*Krauss et al., 2009*). In several other biological systems including neurons and pigment cells (melanophores), competition between long-distance transport along microtubules by kinesins and anchorage/local transport by myosin-V can direct the distribution of cellular cargoes (*Rogers and Gelfand, 1998*; *Wu et al., 1998*; *Kapitein et al., 2013*; *van Bergeijk et al., 2015*; *Pathak et al., 2010*; *Bridgman, 1999*). In the oocyte, kinesin-1 has been proposed to actively remove *osk* mRNA from the cortex (cortical exclusion) by transporting it along cortical-anchored microtubules towards plus-ends of microtubules in the cytoplasm (*Cha et al., 2002*). This cortical exclusion function of kinesin-1 against myosin-V may explain *osk*/Staufen specific anchorage at the posterior pole.

In this study, we propose and directly test a unifying model that explains how *osk*/Staufen is transported in the oocyte and localized at the posterior pole. According to the model, myosin-V directly competes with kinesin-1 to achieve the correct posterior localization of *osk*/Staufen. Specifically, kinesin-1, walking along microtubules, removes *osk*/Staufen from the cortex, while myosin-V, walking along actin filaments that are randomly oriented in the cortical network, dynamically anchors *osk*/Staufen and counteracts kinesin-driven cortical clearance. Kinesin-1 wins this competition at the anterior and lateral regions where cortical microtubule density is higher; myosin-V wins over kinesin-1 at the posterior pole where microtubule tracks are less abundant.

We genetically modulate the motor activities of either kinesin-1 or myosin-V, and find that a proper balance between kinesin-1 and myosin-V activities is essential for *osk*/Staufen posterior localization. Using optogenetic tools we show that cortical microtubule density indeed controls *osk*/Staufen localization. Furthermore, using a synthetic complex that contains both myosin V and kinesin-1 motor domains, we demonstrate that these two motors are not only necessary but also sufficient for posterior localization in the oocyte. Thus, for the first time, we directly demonstrate that direct competition between an actin motor myosin-V and a microtubule motor kinesin-1 controls posterior determinant localization in *Drosophila* oocytes and that the outcome of this competition is determined by the local density of cortical microtubules.

## Results

### Staufen localization is controlled by kinesin-1

If Staufen localization is defined by kinesin-myosin competition, it can be disrupted in a predictable way by modulating the activity of kinesin-1. Knockdown of KHC by RNAi leads to uniform cortical localization of Staufen in the oocyte (*Figure 1—figure supplement 1A–1D*; *Brendza et al., 2000*; *Palacios and St Johnston, 2002*; *Lu et al., 2018*; *Cha et al., 2002*). This indicates that kinesin-1 activity is essential for Staufen cortical exclusion from the anterior and lateral cortex.

To increase kinesin-1 activity in the oocyte, we employed a previously characterized constitutively active KHC mutant, $Khc^{\Delta Hinge2}$, which lacks the flexible hinge region (residues 521–642). As a result, it is unable to fold into an auto-inhibited conformation, resulting in constitutive activation (*Figure 1A*; *Barlan et al., 2013*; *Kelliher et al., 2018*). Staufen staining of heterozygotes carrying one copy of $Khc^{\Delta Hinge2}$ exhibits a more loosened posterior Staufen cap in stage 9 oocytes (*Figure 1—figure supplement 1E, G and I*)), demonstrating that constitutively active kinesin-1 is dominant, and sufficient to partially mislocalize Staufen from the posterior crescent.

Next, we examined Staufen localization in $Khc^{\Delta Hinge2}$ homozygous mutants. As $Khc^{\Delta Hinge2}$ homozygotes do not survive to adulthood (*Kelliher et al., 2018*), $Khc^{\Delta Hinge2}$ homozygous germline clones are induced by heat shock-driven flippase (hs-FLP). Within these $Khc^{\Delta Hinge2}$ homozygous germline clones, Staufen is mislocalized in stage 9 oocytes. Typically, Staufen forms a large aggregate in the middle of the oocyte and a small loose residual cap near the posterior cortex (*Figure 1B–E*). The endogenous deletion of Khc hinge2 region shows a stronger phenotype of Staufen mislocalization than the ectopic expression of $Khc^{\Delta Hinge2}$ in the *Khc* null background from a previous study (*Williams et al., 2014*). Intriguingly, later in development (at stage 10B) normal Staufen distribution

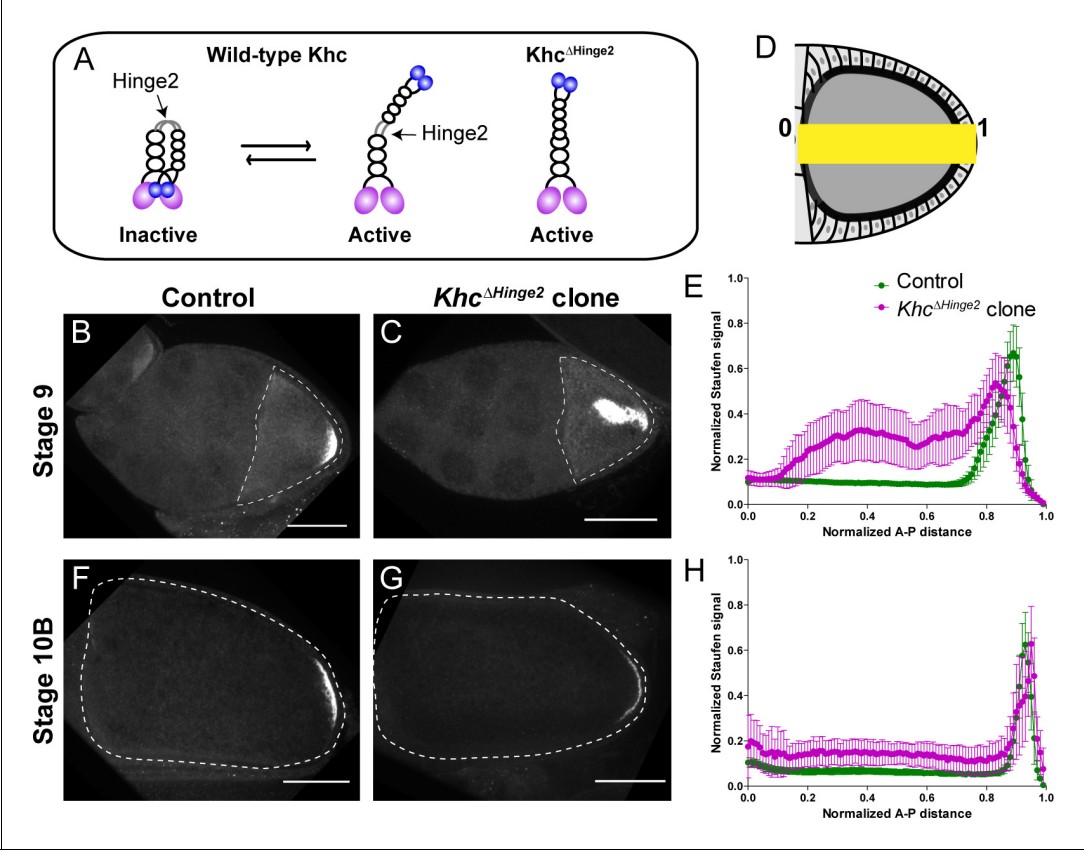

**Figure 1.** Constitutive activity of kinesin-1 leads to cytoplasmic localization of Staufen. (**A**) A cartoon illustration of kinesin-1 heavy chain (KHC) auto-inhibition and constitutive activity of the $Khc^{\Delta Hinge2}$ mutant. (**B–C**) Representative images of Staufen staining in a stage 9 control oocyte (**B**) and a stage 9 $Khc^{\Delta Hinge2}$ homozygous mutant oocyte (**C**). (**D**) An illustration of the measurement of Staufen staining along the A-P axis (See more details in Materials and methods 'Quantification of fluorescence intensity in the oocytes'). (**E**) Normalized Staufen staining signal (average ±95% confidence intervals) along A-P axis in stage 9 control oocytes and $Khc^{\Delta Hinge2}$ homozygous mutant oocytes. Control, N = 38; $Khc^{\Delta Hinge2}$ clone, N = 25. (**F–G**) Representative images of Staufen staining in a stage 10B control oocyte (**F**) and a stage 10B $Khc^{\Delta Hinge2}$ homozygous mutant oocyte (**G**). (**H**) Normalized Staufen staining signal (average ±95% confidence intervals) along A-P axis in stage 10B control oocytes and $Khc^{\Delta Hinge2}$ homozygous mutant oocytes. Control, N = 27; $Khc^{\Delta Hinge2}$ clone, N = 14. Scale bars, 50 μm.

The online version of this article includes the following figure supplement(s) for figure 1:

**Figure supplement 1.** Kinesin-1 activity is essential for correct Staufen localization.

is recovered with the restoration of the posterior cap and clearing of the central cytoplasmic aggregate (*Figure 1F–H*; *Figure 1—figure supplement 1F, H and J*). Previous studies suggest that Osk protein, translated at the posterior pole, functions in a positive feedback mechanism for *osk*/Staufen accumulation in streaming oocytes (*Lu et al., 2018*; *Vanzo and Ephrussi, 2002*). We postulate that enough Osk protein translates from the residual cap, initiating the positive feedback loop, while ooplasmic streaming circulates mislocalized *osk*/Staufen particles to the posterior cap, enhancing Osk localization, resulting the restoration of the posterior crescent.

Together, our data show that *osk*/Staufen posterior localization is sensitive to kinesin-1 activity level.

## Staufen localization is controlled by myosin-V

If *osk*/Staufen localization is indeed controlled by kinesin-myosin competition, a change of myosin-V activity would be predicted to disrupt *osk*/Staufen posterior localization. Previous studies reveal that inhibition of myosin-V by overexpression of the C-terminal myosin-V globular tail (GT) causes Staufen mislocalization to the center of the oocyte (*Krauss et al., 2009*; *Lu et al., 2018*), which is in agreement with our hypothesis that myosin-V anchors *osk*/Staufen at the posterior cortex. In addition, our hypothesis predicts that increased myosin-V activity alters the outcome of the competition between

these two motors, and results in ectopic localization of *osk*/Staufen at the anterior-lateral cortex. Therefore, we decided to disrupt the regulation of myosin-V activity. Similar to kinesin-1, myosin-V is auto-inhibited through its C-terminal cargo-binding GT domain binding to the motor and inhibiting its motor ATPase activity (*Thirumurugan et al., 2006*; *Donovan and Bretscher, 2015*; *Li et al., 2008*; *Figure 2A*). The C-terminus of the coiled-coil 1 region is essential for auto-inhibition as it docks the cargo-binding tail for binding to the motor domain, stabilizing the closed inactive conformation (*Figure 2A*; *Donovan and Bretscher, 2015*; *Li et al., 2008*; *Li et al., 2006*). Deletion of the C-terminus of the coiled-coil 1 region of mouse myosin Va abolishes the inhibitory function of the GT domain and leads to constitutive activity of myosin-Va (*Li et al., 2006*). Based on the homology between *Drosophila* myosin-V and mouse myosin Va, and coiled-coil region predictions, we deleted the 98 residues at the C-terminus of coiled-coil 1 region to create MyoV$^{\Delta 1017\text{-}1114}$ (*Figure 2A*).

We purified this myosin-V deletion mutant (*Figure 2—figure supplement 1A*; *Wang et al., 2000*; *Tóth et al., 2005*) to perform actin gliding assays of myosin-V variants. MyoV$^{\Delta 1017\text{-}1114}$ shows a similar velocity in actin gliding assays compared to wild-type full-length myosin-V (*Figure 2B*), indicating that the deletion does not disrupt myosin-V motor function. Next, we examined whether this deletion leads to constitutive activity by measuring the steady-state ATPase activity of these variants. Wild-type myosin-V displays a low ATPase activity in the absence of $Ca^{2+}$, as it adopts a closed inactive conformation (*Li et al., 2006*; *Wang et al., 2004*), but it is significantly stimulated by $Ca^{2+}$ due to release from the auto-inhibited conformation (*Wang et al., 2004*; *Figure 2C*). In contrast, MyoV$^{\Delta 1017\text{-}1114}$ displays an equivalent and high level of ATPase activity with and without $Ca^{2+}$, suggesting the deletion mutant is constitutively active (*Figure 2D*). Furthermore, we performed negative stain electron microscopy on wild-type myosin-V and myosin-V$^{\Delta 1017\text{-}1114}$. Most wild-type myosin-V molecules at a low salt concentration adopt a closed conformation (*Figure 2E and I*; *Figure 2—figure supplement 1B*), while in high salt concentration they display an open conformation (*Figure 2F*). The closed conformation of wild-type myosin-V observed in low salt closely resembles that previously demonstrated in mammalian myosin Va, indicating that the structural basis for autoinhibition is conserved (*Thirumurugan et al., 2006*; *Liu et al., 2006*). In contrast, myosin-V$^{\Delta 1017\text{-}1114}$ displays an open conformation in both low and high salt conditions (*Figure 2G–H and J*). Measurement of head-junction-head angles shows that the Δ1017–1114 deletion results in a more open conformation than wild-type myosin-V (*Figure 2K*) (see more details in Materials and methods 'Electron Microscopy'). Collectively, the data show that the Δ1017–1114 deletion creates a constitutively active myosin-V.

In order to eliminate interference between the constitutively active mutant and endogenous myosin-V, we generated germline clones of a myosin-V loss-of-function mutant, *didum*[88], in which Myosin-V level is dramatically reduced (*Krauss et al., 2009*). We expressed MyoV$^{\Delta 1017\text{-}1114}$ in *didum*[88] germline clones and find that Staufen staining in these oocytes becomes less restricted at the posterior pole and spreads to the lateral cortex (*Figure 2L–N*). The effect is more pronounced at the anterior half of the lateral cortex (*Figure 2M–N*). Thus, in agreement with the proposed role of myosin-V as an anchor for *osk*/Staufen at the actin cortex, constitutively active myosin-V expands the domain of cortically anchored *osk*/Staufen beyond the posterior pole. Together, these results indicate that *osk*/Staufen cortical anchorage is sensitive to myosin-V activity level.

## Anterior-posterior microtubule gradient is essential for posterior determination

The results above demonstrate that the proper balance of kinesin-1 and myosin-V activities is essential for *osk*/Staufen localization at the posterior pole, consistent with the competition model. In other biological systems, such as pigment cells, local enrichment of F-actin favors myosin-V-dependent anchoring over microtubule-dependent transport (*Wu et al., 1997*). However, within the oocyte, neither actin filaments nor myosin-V are concentrated at the posterior pole (*Krauss et al., 2009*; *Lu et al., 2016*; *Figure 2—figure supplement 1C–1F*). In contrast, the microtubule network in the oocyte displays a clear anterior-posterior gradient, with the lowest density of cortical microtubules at the posterior cortex (*Clark et al., 1997*; *Nashchekin et al., 2016*; *Zimyanin et al., 2008*). This anterior-posterior (A-P) microtubule gradient is revealed by microtubule labeling with germline-specific expressed EMTB-3XTagRFP (*Faire et al., 1999; Figure 3A*), and a GFP-labeled endogenous MAP, Jupiter-GFP (*Morin et al., 2001*; *Karpova et al., 2006*; *Figure 3—figure supplement 1A*). This gradient is a result of reduced accumulation of the microtubule minus-end binding protein,

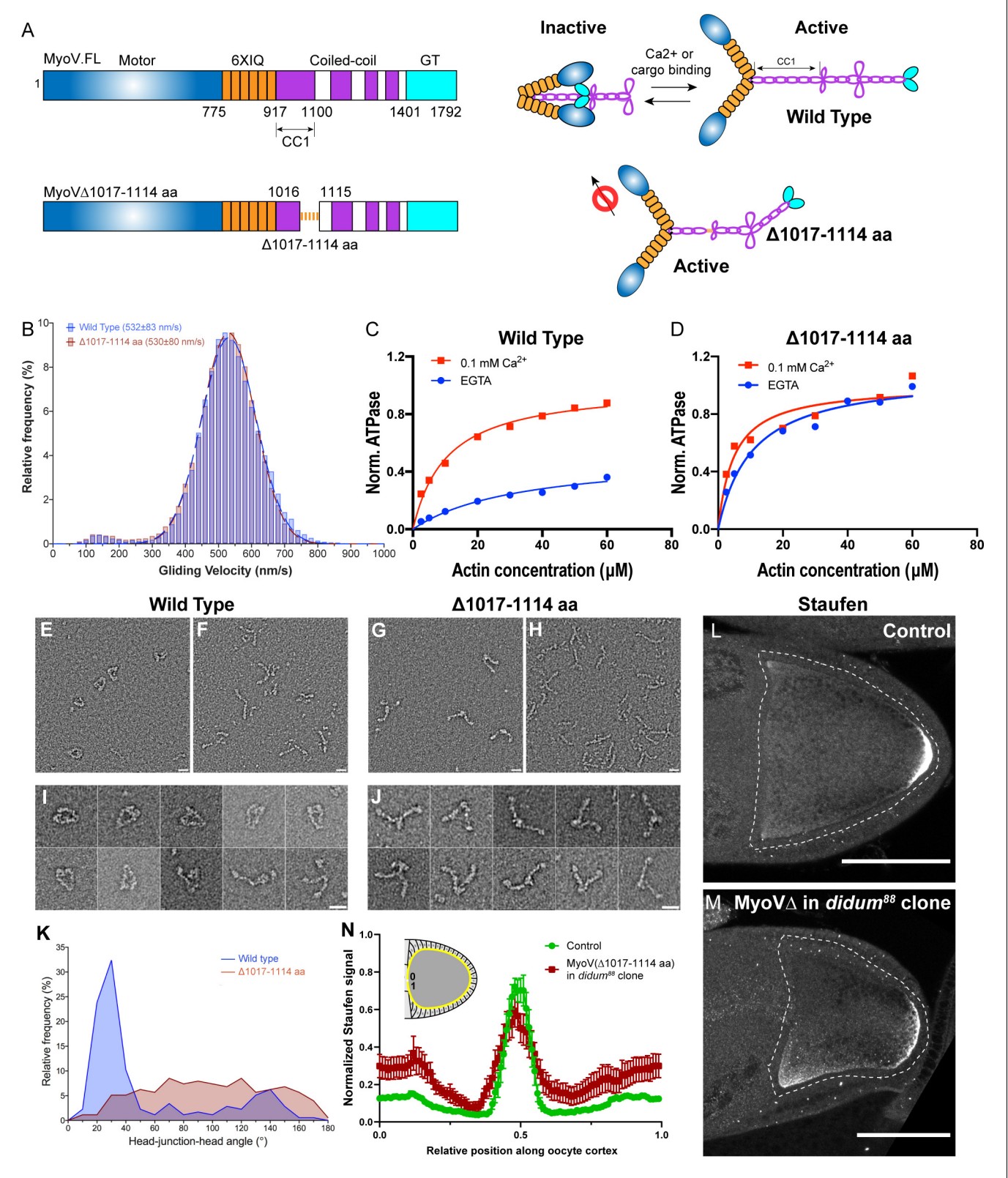

**Figure 2.** Constitutive activity of myosin-V leads to a broader cortical localization of Staufen. (**A**) A schematic illustration of myosin-V domains and auto-inhibition. Globular tail (GT) causes auto-inhibition conformation of myosin-V by interacting with motor domain. This auto-inhibition can be relieved by cargo binding to the GT domain. The inhibitory function of GT domain requires the C-terminal region of the coiled-coil 1 region. (**B**) Gliding velocities

*Figure 2 continued on next page*

*Figure 2 continued*

of wild-type myosin-V (532 ± 83 nm/s, Mean ±95% Confidence Intervals) and myosin-VΔ1017–1114 aa (530 ± 80 nm/s, Mean ±95% Confidence Intervals). (C–D) Steady-state ATPase activity analysis of wild-type myosin-V (C) and myosin-VΔ1017–1114 aa (D). (E–J) Representative electron microscopic images of wild-type myosin-V (E–F, I) and myosin-VΔ1017–1114 aa (G–H, J). (E, G) myosin-V in conditions which favor inhibited conformation (50 mM NaCl, 0.1 mM ATP, 0.1% glutaraldehyde); (F, H) myosin-V in high salt conditions (500 mM NaCl). (I–J) Individual examples of the wild-type myosin-V (I) and myosin-VΔ1017–1114 aa (J) (50 mM NaCl, 0.1 mM ATP, 0.1% glutaraldehyde). Scale bars, 20 nm. (K) Quantification of the head-junction-head angle of wild-type myosin-V and myosin-VΔ1017–1114 aa. (L–M) Staufen localization in control (L) and myosin-VΔ1017–1114 aa-expressing *didum*[88] mutant (M) oocytes. (N) Normalized Staufen staining signal (average ± SEM) along oocyte cortex (illustrated as the yellow line in the inset) in stage 9 control oocytes and myosin-VΔ1017–1114 aa-expressing *didum*[88] mutant oocytes. Control, N = 16; myosin-VΔ1017–1114 expression in *didum*[88] mutant clone, N = 14. Scale bars, 50 μm.

The online version of this article includes the following figure supplement(s) for figure 2:

**Figure supplement 1.** Myosin-V activity is essential for proper Staufen localization.

Patronin, at the posterior pole (*Figure 3B–C*; *Nashchekin et al., 2016*). We hypothesize that the cortical microtubule density is the key factor that determines the outcome of the kinesin-myosin competition.

To test this hypothesis, we first used genetic tools to disrupt the microtubule gradient in oocytes. Germline overexpression of a depolymerizing kinesin, kinesin 13 (Klp10A in *Drosophila*) (*Goshima and Vale, 2005*; *Mennella et al., 2005*), leads to complete depolymerization of microtubules in the oocyte as well in the nurse cells, while the somatic follicle cells in these ovaries still have intact microtubules (*Figure 3E–E'*, compared to *Figure 3D–D'*; *Figure 3—figure supplement 1B–C' and E*). This lack of microtubules in the oocyte results in a failure of Staufen enrichment at the posterior pole; instead it is uniformly localized along the entire oocyte cortex (*Figure 3G–H and J*).

Conversely, we increased the density of cortical microtubules in the oocyte by overexpressing the microtubule minus-end binding protein, Patronin. Patronin stabilizes microtubules and prevents minus-end depolymerization by Klp10A (*Goodwin and Vale, 2010*). Overexpressing Patronin significantly increases microtubule density in the oocyte (*Figure 3F–F'*; *Figure 3—figure supplement 1D–1E*). This increase in microtubule density causes Staufen to be excluded from the entire cortex. Instead, it forms aggregates in the center of the oocyte where most microtubules plus-ends are directed (*Figure 3G, I and K*).

Together, these data demonstrate that *osk*/Staufen localization is controlled by the microtubule density in the oocyte. Decreased density of cortical microtubules favors myosin-V resulting in uniform cortical localization of *osk*/Staufen, whereas increased microtubule density favors kinesin-1 resulting in exclusion of *osk*/Staufen from the cortex.

## Manipulation of local microtubule density changes Staufen localization

Having established that the global manipulations of the microtubule gradient disrupt the posterior localization of *osk*/Staufen, we decided to directly test whether a local decrease of the cortical microtubule density is sufficient for cortical accumulation of Staufen. We employed optogenetic tools to locally recruit kinesin-13/Klp10A, which depolymerizes MTs (*Figure 3E*), to the actin cortex to manipulate the local microtubule density in the oocyte (*Figure 4A*). Using an improved light inducible dimer (iLID) (*Guntas et al., 2015*; *Adikes et al., 2018*), GFP-tagged Klp10A-SspB is recruited to the actin-rich cortex within seconds upon global blue light exposure (488 nm) (*Figure 4—Video 1*). We combined the microtubule labeling EMTB-3XTagRFP with Klp10A local recruitment, and found that locally recruiting Klp10A to the lateral cortex significantly decreases the EMTB-3XTagRFP signal (*Figure 4 —Videos 2* and *3*; *Figure 4—figure supplement 1*). We then combined Klp10A recruitment with RFP-tagged Staufen in late stage 8 oocytes prior to the formation of a compact posterior cap of Staufen with a pool of free Staufen particles remaining in the cytoplasm that are potentially available for dynamic anchorage. We locally recruited Klp10A to the lateral cortex, an area of the oocyte that normally has high microtubule density (*Figure 3A*) and no cortical Staufen (*Figure 3G*). Recruitment of Klp10A-SspB to the lateral cortex leads to a dramatic accumulation of RFP-Staufen in the stimulated area (*Figure 4B–D*). Over the time of stimulation, the RFP-Staufen progressively accumulated at the cortex (*Figure 4C and E*). Remarkably, this ectopic recruitment of Staufen to the region of reduced microtubule density is reversible; the Staufen signal decreased to control levels after removal of blue light. Furthermore, this ectopic recruitment is repeatable. We are

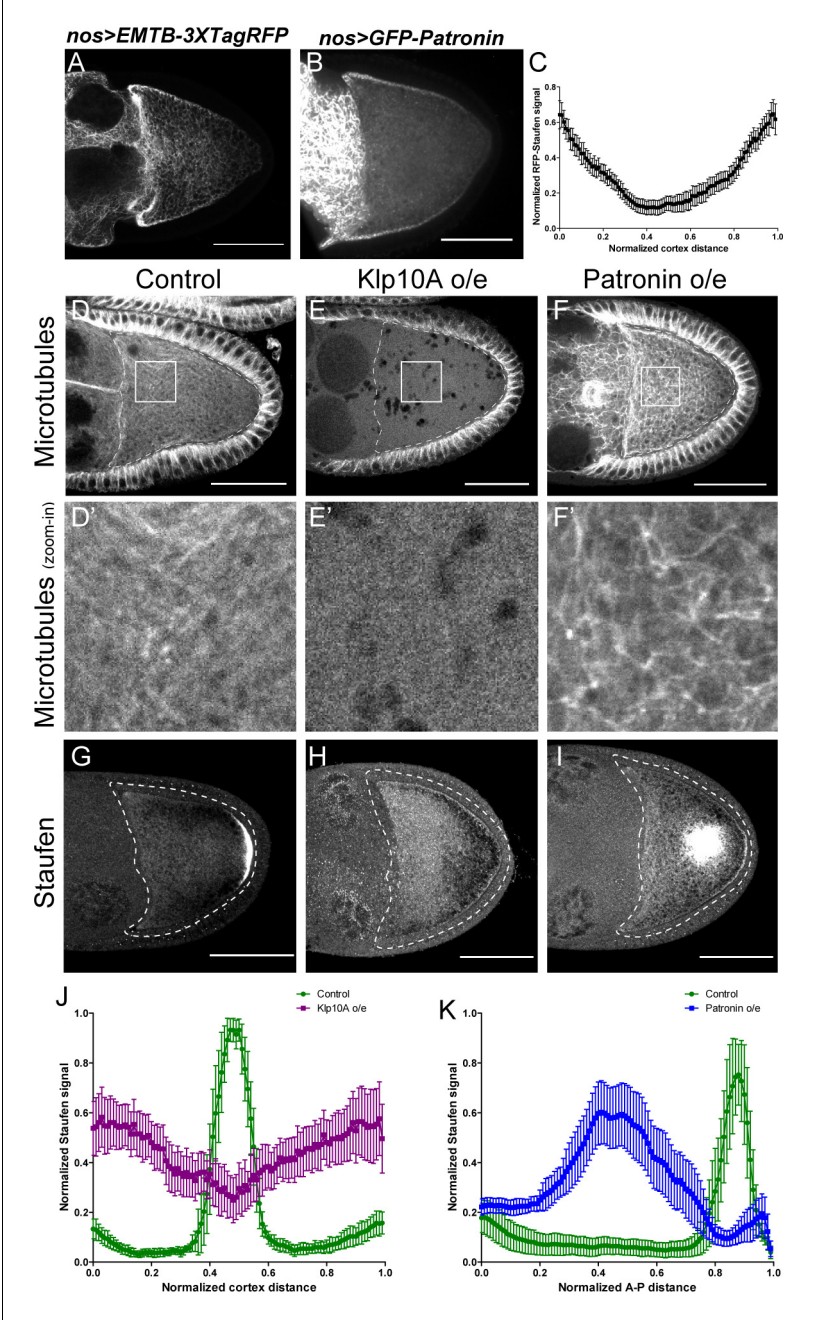

**Figure 3.** Anterior-posterior microtubule gradient is essential for correct Staufen posterior localization. (A–B) EMTB-3XTagRFP (**A**) and GFP-Patronin (**B**) localization in stage 9 oocytes. (**C**) Normalized GFP-Patronin signal (average ±95% confidence intervals) along the oocyte cortex in stage 9 oocytes (as shown in *Figure 1—figure supplement 1C*; see more details in Materials and methods 'Quantification of fluorescence intensity in the oocytes'; N = 42). (D–F) Representative images of microtubule staining (using 647nm-conjugated nanobody against tubulin after extraction) in stage 9 control (**D**), Klp10A-overexpressing (**E**), and Patronin-overexpressing (**F**) oocytes. (D'–F') Zoom-ins of the boxed areas in (D–F). (G–I) Representative images of Staufen staining in stage 9 control (**G**), Klp10A-overexpressing (**H**), and Patronin-overexpressing (**I**) oocytes. (**J**) Normalized Staufen signal (average ±95% confidence intervals) along the oocyte cortex in stage 9 control and Klp10A-overexpressing oocytes (as shown in *Figure 1—figure supplement 1C*). Control, N = 13; Klp10A overexpression, N = 20. (**K**) Normalized Staufen signal (average ±95% confidence intervals) along the A-P axis in stage 9 control and Patronin-overexpressing oocytes (as shown in *Figure 1D*). Control, N = 12; Patronin overexpression, N = 30. Scale bars, 50 μm.

*Figure 3 continued on next page*

*Figure 3 continued*

The online version of this article includes the following figure supplement(s) for figure 3:

**Figure supplement 1.** Disruptions of anterior-posterior microtubule gradient using genetic mutants.

able to induce the recruitment either at a single spot of an oocyte (*Figure 4 — Videos 4* and *5*) or at different spots (*Figure 4 —Video 6*). In contrast, in the samples expressing LifeAct-SsrA only (No Klp10A-SspB), no increase of RFP-Staufen is observed after the identical light treatment, indicating the effect of Staufen-accumulation is specific to Klp10A recruitment (*Figure 4D*).

To ensure that the local recruitment effect we observed is explained by Klp10A-dependent microtubule depolymerization (instead of Klp10A directly recruiting Staufen), we treated the flies with colcemid to globally depolymerize microtubules. After colcemid treatment, no visible microtubules remain at the cortex (*Figure 4—figure supplement 2*). Under these conditions, Klp10A recruitment at the actin cortex does not cause Staufen recruitment (*Figure 4D*).

Collectively, these results demonstrate that local microtubule density is the key determinant of *osk*/Staufen localization by tipping the balance of kinesin-myosin competition. Lower microtubule density favors cortical anchorage by myosin-V, while higher microtubule density favors cortical exclusion by kinesin-1.

## Competition between kinesin-1 and myosin-V is sufficient for posterior localization

We further tested whether the competition between kinesin-1 and myosin-V is sufficient to drive posterior accumulation of key polarity determinants in the oocyte. In order to avoid any effects of cargo interactions, we designed a minimal artificial system and used a rapalog-dependent dimerization system (*Kapitein et al., 2013*; *DeRose et al., 2013*) to induce complex formation between constitutively active dimers of kinesin-1 heads (KHC576) (*Ally et al., 2009*) and myosin-V heads (MyoVHMM) (*Tóth et al., 2005*; *Rosenfeld and Lee Sweeney, 2004*; *Figure 5A*).

In the absence of rapalog, KHC576-TagRFP is distributed mostly in cytoplasm, consistent with the fact that the plus-ends of the cortical microtubules are directed away from the cortex (*Figure 5B and C–C′*). GFP-tagged MyoVHMM in these oocytes is uniformly localized at the actin cortex, consistent with the fact that it is bound to F-actin (*Figure 5C*). Upon addition of rapalog, the distribution of KHC576 changes dramatically; it mostly accumulates at the posterior pole, resembling the localization of *osk*/Staufen (*Figure 5B and D–D′*). Some cortical localization of MyoVHMM remains at the anterior and lateral cortex, likely because it was expressed in the oocyte in excess of KHC576.

As a control, we repeated the rapalog treatment after microtubule depolymerization by colcemid. In the absence of microtubules, the dimerized KHC576-MyoVHMM shows no accumulation at the posterior pole (*Figure 5B*).

In summary, our rapalog experiments reveal that a synthetic protein complex containing just two active motors, kinesin-1 and myosin-V, exhibits microtubule-dependent posterior localization in *Drosophila* oocytes. This dimerized complex is removed by kinesin-1 from the anterior-lateral cortex that has more cortical microtubules and trapped by myosin-V at the posterior cortex where the cortical microtubule density is much lower (*Figure 5E*).

## Discussion

It is well established that kinesin-1 is essential for localization of *osk*/Staufen particles at the posterior pole of the *Drosophila* oocyte. However, it remained unclear how the compact posterior cap is anchored and retained over time. Cortical F-actin remodeling and Myosin-V, as well as the Arp1 subunit of the dynactin complex, have been all implicated in the *osk*/Staufen cortical localization (*Krauss et al., 2009*; *Nieuwburg et al., 2017*; *Tanaka et al., 2011*; *Tanaka and Nakamura, 2011*). In this study, we combined genetic and optogenetic tools to demonstrate that direct competition between two motors, kinesin-1 and myosin-V, ensures the posterior anchorage of *osk*/Staufen. Notably, we demonstrate that the outcome of the competition is primarily determined by the density of cortical microtubules. High microtubule density at the anterior and lateral cortex favors kinesin-driven *osk*/Staufen cortical exclusion, while low microtubule density at the posterior pole favors

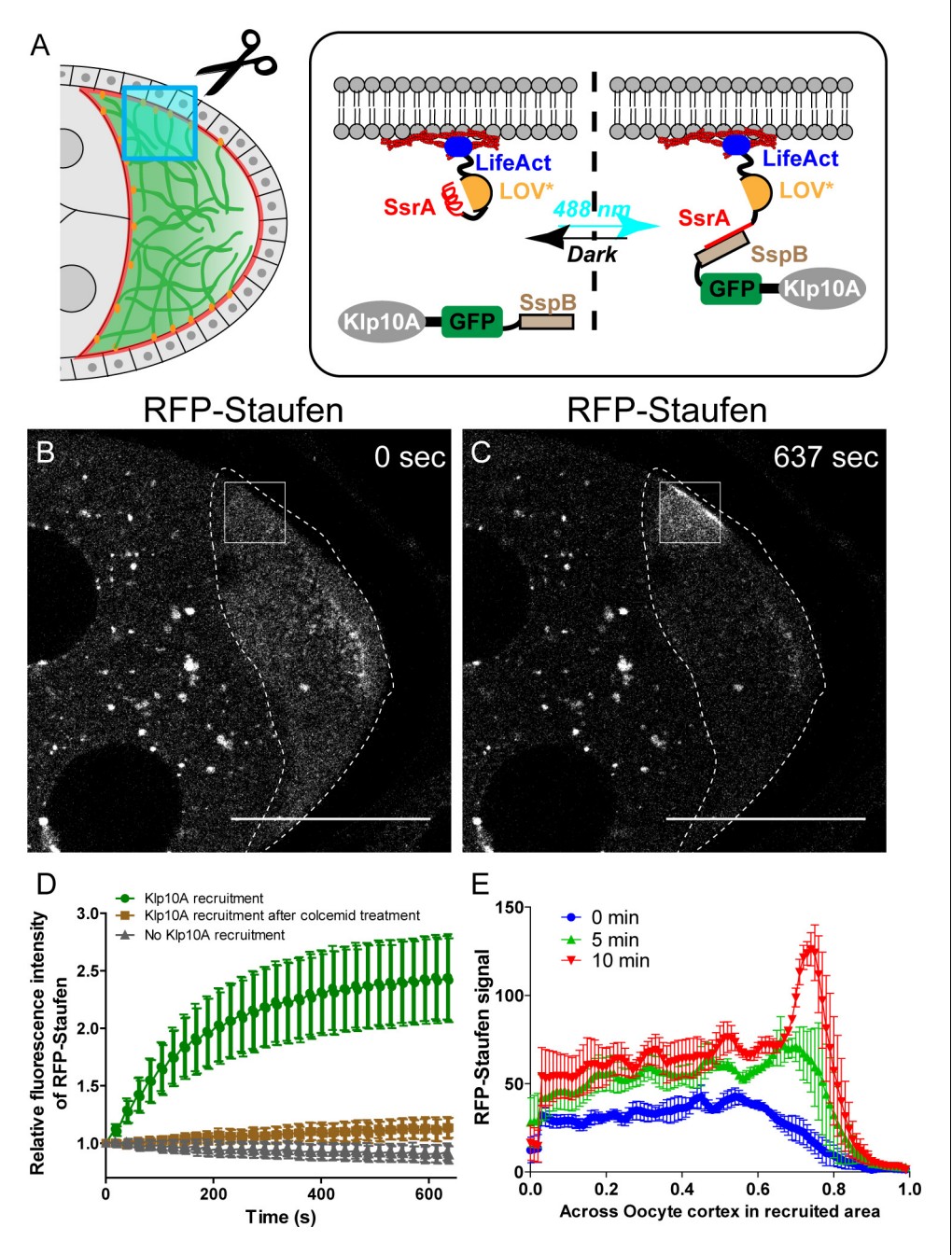

**Figure 4.** Local depolymerization of microtubules causes ectopic Staufen accumulation on the lateral cortex. (**A**) iLID-mediated optogenetic strategy: manipulation of cortical microtubule density through recruiting a microtubule depolymerizer kinesin-13/Klp10A to LifeAct-labeled actin cortex by blue light. (**B–C**) Representative images of RFP-Staufen before (**B**) and after (**C**) local recruitment of Klp10A (square box) in late stage 8 oocytes. Scale bars, 50 μm. (**D**) Quantification of RFP-Staufen (average ±95% confidence intervals) in the Klp10A-recruited region (see more details in Material and methods 'Optogenetic recruitment'). The RFP-Staufen fluorescence intensity in the recruitment region was normalized according to the 'Before' image. Klp10A recruitment, N = 25; Klp10A recruitment after colcemid treatment, N = 24; No Klp10A recruitment, N = 27. (**E**) RFP-Staufen signal (average ± SEM) was measured along a line (~6 μm width) across the cortex within the recruited area at 0 min, 5 min and 10 min of recruitment. Distance across the cortex was normalized (N = 4; see more details in 'Quantification of fluorescence intensity in the oocytes').

The online version of this article includes the following figure supplement(s) for figure 4:

**Figure supplement 1.** Decrease of cortical microtubule density using optogenetic tools.

*Figure 4 continued on next page*

*Figure 4 continued*
**Figure supplement 2.** Colcemid treatment causes microtubule depolymerization.

myosin-driven cortical retention. Therefore, the kinesin-myosin competition and cortical microtubule density together determine the initial accumulation of *osk* mRNA at the posterior pole (*Figure 5E*).

## Kinesin-driven delivery versus kinesin-driven cortical exclusion

The cortical exclusion model was first proposed after uniform cortical localization of *osk* mRNA was observed in the kinesin-null oocytes (*Cha et al., 2002*). In agreement with this model, we show that constitutively active kinesin-1 causes *osk*/Staufen mislocalization in the cytoplasm of the oocyte (*Figure 1*), whereas reducing microtubule density at the lateral cortex leads to ectopic accumulation of Staufen at the cortex (*Figure 4*). These data support the model that kinesin-driven cortical exclusion along cortically-attached microtubules plays an essential role in restricting *osk*/Staufen to the posterior pole.

Previously, several groups have proposed that kinesin-1 transports *osk*/Staufen particles along slightly biased cortical microtubules, resulting in net movement of *osk*/Staufen from the anterior side to the posterior pole in stage 8–9 oocytes (*Palacios and St Johnston, 2002*; *Lu et al., 2018*; *Zimyanin et al., 2008*; *Parton et al., 2011*). In fact, kinesin-driven cortical exclusion and kinesin-driven transport towards the posterior pole are not mutually exclusive; they describe the same event of *osk*/Staufen movement. Within the oocyte, cortical microtubules are anchored to the cortex by their minus-ends while their plus-ends face towards the cytoplasm. Due to the anterior-posterior gradient of cortical microtubule density, more microtubule plus ends are oriented towards the posterior pole (*Nashchekin et al., 2016*; *Zimyanin et al., 2008*; *Parton et al., 2011*; *Khuc Trong et al., 2015*). Thus, kinesin-1-driven transport along microtubules is a prerequisite for kinesin-1-driven cortical exclusion. Cortical exclusion of *osk*/Staufen by kinesin-1 results in biased transport of *osk*/Staufen towards the posterior pole (*Figure 5E*).

## Kinesin-myosin tug-of-war, oocyte style

This kinesin-myosin competition model is suggested by genetic interaction data from a previous study. Specifically, increasing KHC dosage enhances *osk*/Staufen mislocalization phenotypes in myosin-V loss-of-function mutants, while reducing KHC dosage by half partially suppresses myosin loss-of-function phenotypes (*Krauss et al., 2009*). Furthermore, double *MyoV* and *Khc* mutant clones have diffuse cytoplasmic localization of *osk* mRNA, compared to uniform cortical localization of *osk*

mRNA in *Khc* single mutant clones (*Krauss et al., 2009*). These data strongly imply that in the absence of kinesin-1, myosin-V promiscuously anchors *osk*/Staufen everywhere in the cortex. In this study, we manipulate the activity of either kinesin-1 or myosin-V, and find that proper balance between the activities of these two motors is critical for correct *osk*/Staufen localization, supporting the model in which kinesin-myosin competition is key to the correct posterior determination in the *Drosophila* oocyte.

The competition between kinesin-1 and myosin-V we described here is not the first example of such a mechanism for cargo transport and localization. For instance, myosin-V opposes microtubule-dependent transport and provides a dynamic anchor for melanosomes (*Rogers and Gelfand, 1998*; *Wu et al., 1998*; *Wu et al., 1997*), peroxisomes (*Kapitein et al., 2013*; *van Bergeijk et al., 2015*), recycling endosomes (*van Bergeijk et al., 2015*), mitochondria

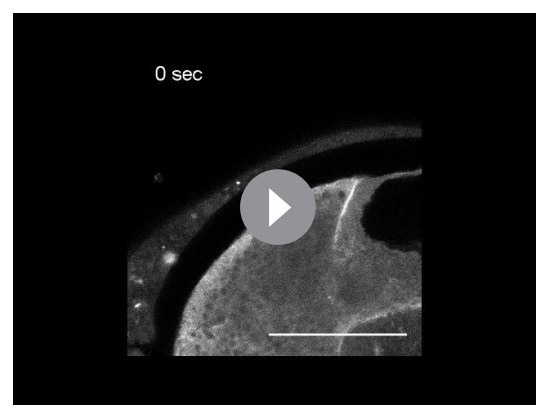

**Video 1.** Global recruitment of Klp10A-GFP-SspB to actin cortex by LifeAct-SsrA. 488 nm laser was used to induce the recruitment between SsrA and SspB (shown as the blue box in the upper-right corner). Scale bar, 50 µm.
https://elifesciences.org/articles/54216#video1

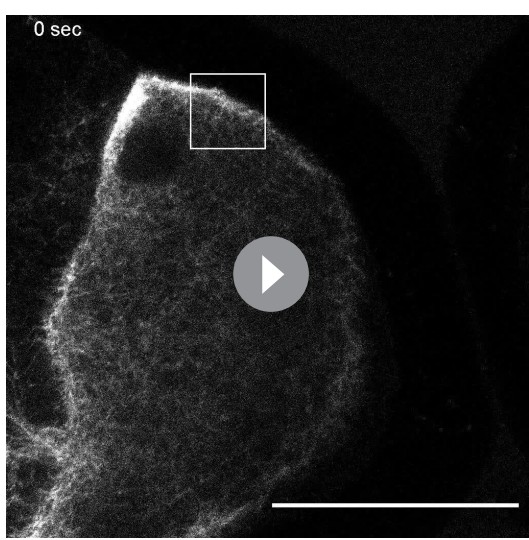

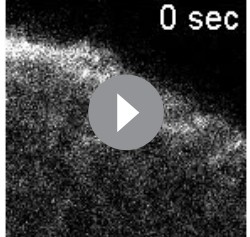

**Video 3.** Local recruitment of Klp10A-GFP-SspB causes decrease of microtubule signal. Microtubules were labeled by EMTB-3XTagRFP. Zoom-in area of the local microtubule signal as shown in *Video 2* (outlined by the white box).

https://elifesciences.org/articles/54216#video3

**Video 2.** Local recruitment of Klp10A-GFP-SspB causes decrease of microtubule signal. Microtubules were labeled with EMTB-3XTagRFP. The white box indicates the local recruitment area. The 488 nm light recruitment parameters used in EMTB-3XTagRFP samples were identical to the ones applied to RFP-Staufen samples. Scale bar, 50 µm.

https://elifesciences.org/articles/54216#video2

(*Pathak et al., 2010*), and synaptic vesicles (*Bridgman, 1999*) at sites of local accumulation of F-actin. At these sites, the abundance of F-actin tracks provides an upper hand for myosin-V to win the tug-of-wars over microtubule motors. Kinesin-myosin competition appears to be an evolutionarily conserved mechanism to allow flexible refinement and/or error correction, as motors constantly undergo reversible binding and releasing activity on cytoskeletal filaments.

In the oocytes, the machinery responsible for *osk*/Staufen localization contains the same basic building blocks; however, unlike the other systems, the outcome of the competition is not determined by actin filament density, as F-actin density is uniform along the oocyte cortex (*Figure 2—figure supplement 1*). Instead, the outcome of this competition is decided by abundance of microtubule tracks. Higher microtubule density at the anterior and lateral cortex favors kinesin-mediated cortical removal of *osk*/Staufen, while scarcity of microtubule tracks at the posterior pole favors myosin-V-dependent anchoring. To confirm this model, we used optogenetic tools to recruit a microtubule-depolymerizing kinesin, kinesin-13/Klp10A, to actin cortex, and thus locally modulate cortical microtubule density. Locally decreasing cortical microtubule density causes ectopic accumulation of Staufen at the cortex. The loss of microtubules prevents kinesin-driven cortical exclusion, which allows myosin-V to win the competition and form a patch of cortically localized Staufen. This recruitment of Staufen is reversible and repeatable, indicating this kinesin-myosin competition is continuous, and the outcome of this never-ending battle is decided by the local microtubule concentration.

## Competition between kinesin-1 and myosin-V is sufficient for initial anchoring at the posterior pole

Previously, synthetic motor domains of a plus-end motor, kinesin-1, and a minus-end motor, kinesin-14/Nod, were used to label overall microtubule polarity in *Drosophila* oocytes and neurons (Kin:βGal and Nod:βGal) (*Clark et al., 1994*; *Clark et al., 1997*). As myosin-V is

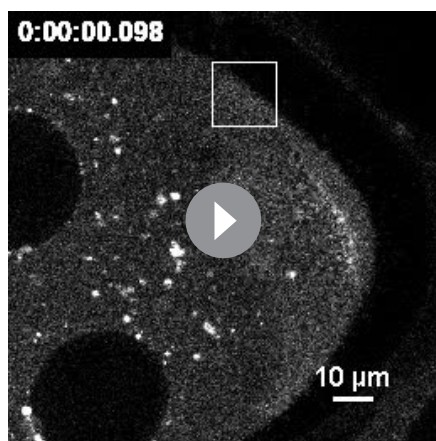

**Video 4.** Local recruitment of Klp10A-GFP-SspB to F-actin results in RFP-Staufen accumulation at the local cortex. The white box indicates the local recruitment area. Scale bar, 10 µm.

https://elifesciences.org/articles/54216#video4

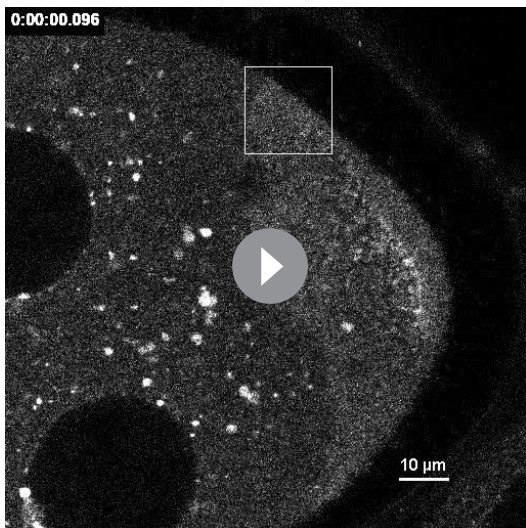

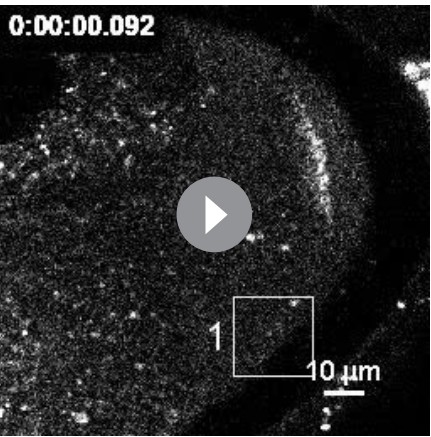

**Video 5.** Accumulation of RFP-Staufen at the cortex by Klp10A recruitment is reversible and repeatable. The 2nd recruitment (outlined by the white box) was performed in the same sample as *Video 4* after resting in complete dark for 30 min. Scale bar, 10 μm.
https://elifesciences.org/articles/54216#video5

**Video 6.** Local recruitment of Klp10A-GFP-SspB results in RFP-Staufen accumulation in multiple sites of an oocyte. The white boxes indicate the local recruitment areas. Three recruitments were performed sequentially (shown as '1', '2' and '3'). Scale bar, 10 μm.
https://elifesciences.org/articles/54216#video6

essential for *osk*/Staufen localization, in this study we expressed two synthetic motor constructs (KHC576 and MyoVHMM) in the oocyte, and induced their dimerization using a rapalog-dependent dimerization system. Dimerized motors accumulate at the posterior pole, highly resembling the *osk*/Staufen localization (*Figure 5*). This posterior accumulation is dependent on the anterior-posterior microtubule gradient; dimerized motors fail to localize at the posterior pole after microtubule depolymerization (*Figure 5*). Together, using dimerized synthetic motors, we demonstrate that direct competition (without any cargo binding) between a microtubule motor, kinesin-1, and an actin motor, myosin-V, is sufficient for initial posterior localization in a *Drosophila* oocyte.

In summary, we have elucidated the anchorage mechanism for initial posterior localization of *osk*/Staufen during mid-oogenesis. Kinesin-1 competes with myosin-V to control *osk*/Staufen localization. The outcome of this kinesin-myosin competition is primarily determined by cortical microtubule density. Higher microtubule density at anterior-lateral cortex allows kinesin-1 to win and cortically exclude *osk*/Staufen, while lower microtubule density at posterior pole favors myosin-V-mediated anchorage at the cortex. Together, two cytoskeletal components (microtubules and F-actin) and two molecular motors (kinesin-1 and myosin-V) govern the posterior determination for future *Drosophila* embryos.

# Materials and methods

## Key resources table

| Reagent type (species) or resource | Designation | Source or reference | Identifiers | Additional information |
|---|---|---|---|---|
| Gene (*Drosophila melanogaster*) | Staufen (Stau) | DOI: 10.1083/jcb.201709174 | FBgn0003520; CG5753 | |
| Gene (*Drosophila melanogaster*) | kinesin heavy chain (khc) | Isabel Palacios lab; DOI: 10.1242/dev.097592 | FBgn0001308; CG7765 | |
| Gene (*Drosophila melanogaster*) | myosin-V (didum) | DOI: 10.1074/jbc.M505209200 | FBgn0261397; CG2146 | |
| Gene (*Drosophila melanogaster*) | oskar (osk) | DOI: 10.1083/jcb.201709174 | FBgn0003015; CG10901 | |

*Continued on next page*

*Continued*

| Reagent type (species) or resource | Designation | Source or reference | Identifiers | Additional information |
|---|---|---|---|---|
| Gene (*Drosophila melanogaster*) | Klp10A | DGRC cDNA clone, LD29208 | FBgn0030268; CG1453 | |
| Gene (*Drosophila melanogaster*) | Patronin | Uri Abdu lab (Ben Gurion University) | FBgn0263197; CG33130 | |
| Genetic reagent (*Drosophila melanogaster*) | khcΔhinge2 | Jill Wildonger lab (University of Wisconsin-Madison); DOI: 10.1083/jcb.201708096 | | |
| Genetic reagent (*Drosophila melanogaster*) | Khc-RNAi (GL00330, Valium22, attP2) | Bloomington stock center | BDSC: #35409 (GL00330) | |
| Genetic reagent (*Drosophila melanogaster*) | maternal α-tubulin 67C-Gal4-VP16[V2H] | Bloomington stock center | BDSC: #7062 | |
| Genetic reagent (*Drosophila melanogaster*) | maternal α-tubulin 67C-Gal4-VP16[V37] | Bloomington stock center | BDSC: #7063 | |
| Genetic reagent (*Drosophila melanogaster*) | nos-Gal4-VP16 (III) | Edwin Ferguson lab (University of Chicago); DOI: 10.1016/s0960-9822(98)70091-0 | | |
| Genetic reagent (*Drosophila melanogaster*) | hs-Flp[12]; Sco/CyO | Bloomington stock center | BDSC: #1929 | |
| Genetic reagent (*Drosophila melanogaster*) | FRTG13, Ubi-GFP.nls. 2R1, Ubi-GFP.2R2 | Bloomington stock center | BDSC: #5826 | |
| Genetic reagent (*Drosophila melanogaster*) | FRTG13 | Bloomington stock center | BDSC: #1956 | |
| Genetic reagent (*Drosophila melanogaster*) | UASp-MyoV.FL-GFP (III) | Anne Ephrussi lab (EMBL); DOI: 10.1016/j.cub.2009.04.062 | | |
| Genetic reagent (*Drosophila melanogaster*) | FRT42B didum[88] | Anne Ephrussi lab (EMBL); DOI: 10.1016/j.cub.2009.04.062 | | |
| Genetic reagent (*Drosophila melanogaster*) | UASp-MyoVΔ1017–1114 aa-FLAG (attP64) | Generated in this study | | |
| Genetic reagent (*Drosophila melanogaster*) | vasa-Gal4 (III) | Allan Spradling lab (Carnegie Institution for Science); DOI: 10.1534/genetics.118.300874 | | |
| Genetic reagent (*Drosophila melanogaster*) | UASp-EMTB-3XTagRFP (III) | Generated in this study | | |
| Genetic reagent (*Drosophila melanogaster*) | UASp-GFP-Patronin (II) | Uri Abdu lab (Ben Gurion University) | | |
| Genetic reagent (*Drosophila melanogaster*) | UASp-Klp10A-GFP-SspB (II) | Generated in this study | | |
| Genetic reagent (*Drosophila melanogaster*) | UASp-LifeAct-HA-SsrA (II) | Generated in this study | | |
| Genetic reagent (*Drosophila melanogaster*) | maternal α-tubulin67 -RFP-Staufen (III) | Daniel St Johnston lab (University of Cambridge); DOI: 10.1083/jcb.201103160 | | |

*Continued on next page*

*Continued*

| Reagent type (species) or resource | Designation | Source or reference | Identifiers | Additional information |
|---|---|---|---|---|
| Genetic reagent (*Drosophila melanogaster*) | vasa-Gal4 (II) | Yukiko Yamashita lab (University of Michigan) | | |
| Genetic reagent (*Drosophila melanogaster*) | UASp-KHC576-TagRFP-FKBP (III) | Generated in this study | | |
| Genetic reagent (*Drosophila melanogaster*) | UASp-MyoVHMM-GFP-FRB (II) | Generated in this study | | |
| Genetic reagent (*Drosophila melanogaster*) | Jupiter-GFP | Yale GFP Protein Trap Database (ZCL2183); DOI: 10.1073/pnas.261408198 | | |
| Genetic reagent (*Drosophila melanogaster*) | UASp-tdEOS2-alpha-tubulin84B (II) | DOI: 10.1016/j.cub.2013.04.050 | BDSC: #51313 | |
| Cell line (*Spodoptera frugiperda*) | Sf9 cells | Invitrogen(Thermo fisher); DOI: 10.1074/jbc.M113.499848; DOI: 10.7554/eLife.32871 | | Maintained in J. Sellers lab; used for recombinant baculovirus expression |
| Antibody | Mouse anti-*Drosophila* Staufen antibody | Chris Q. Doe lab (University of Oregon); DOI: 10.1073/pnas.1522424113; DOI: 10.1083/jcb.201709174 | | IF (1:50) |
| Antibody | Fluorescein (FITC) AffiniPure Goat Anti-Mouse IgG (H+L) | Jackson ImmunoResearch | Cat# 115-095-062 | IF (1:100) |
| Antibody | Rhodamine (TRITC)-AffiniPure Goat Anti-Mouse IgG (H+L) | Jackson ImmunoResearch | Cat# 115-025-003 | IF (1:100) |
| Antibody | 647-conjugated anti-tubulin nanobody (llamas/*E. coli*) | Helge Ewers lab (Freie Universität Berlin); DOI: 10.1038/ncomms8933 | | IF (1:50) |
| Chemical compound, drug | Rhodamine-labeled phalloidin | Thermo Fisher Scientific | Cat # R415 | IF (1:5000) |
| Chemical compound, drug | Colcemid | Cayman Chemical | Item No. 15364 | 200 µM |
| Chemical compound, drug | Rapalog (A/C Heterodimerizer) | Clontech/Takara | Cat# 635055 | 10 µM |
| Software, algorithm | A custom MatLab program that normalized the distance and fluorescent signal in the plot profiles | David Kirchenbüechler (CAM, Northwestern University); DOI: 10.1083/jcb.201709174 | | |

## Plasmid constructs

pFB1-MyoV(Δ1017–1114 aa)-FLAG and pUASp-MyoV(Δ1017–1114 aa)-FLAG: Full-length myosin-V heavy chain (1–1792 residues) with C-terminal FLAG was inserted into pFB1 vector by BamHI (5')/ XbaI (3'). pFB1-MyoVΔ1017–1114 aa deletion was generated by replacing the region of pFB1-MyoV. FL with synthesized oligos by PmeI (5')/NsiI (3'). MyoVΔ1017–1114-FLAG was then subcloned into pUASp-attB vector by BamHI (5')/XbaI (3'). pUASp-Klp10A-GFP-SSpB: Klp10A CDS was amplified by PCR from Klp10A cDNA clone (LD29208, DGRC) and inserted into pUbi-GFP-SspB construct by NheI/AgeI. Klp10A-GFP-SspB fragment was then amplified by PCR and inserted into pUASp by KpnI (5')/SpeI (3'). pUASp-LifeAct-HA-SsrA: LifeAct and HA-SsrA were inserted into pUASp vector by KpnI(5')/SpeI(3') and BamHI(5')/XbaI(3'), respectively. pUASp-EMTB-3XTagRFP: TagRFP was amplified by PCR and 3 copies of TagRFP were cloned into pUASp-attB vector by SpeI(5')/BamHI(3') using In-Fusion cloning (Takara); human Ensconsin MT binding domain (18–283 aa) was amplified by PCR and inserted into pUASp-attB-3XTagRFP by NotI(5')/SpeI(3'). pUASp-KHC576-TagRFP-FKBP:

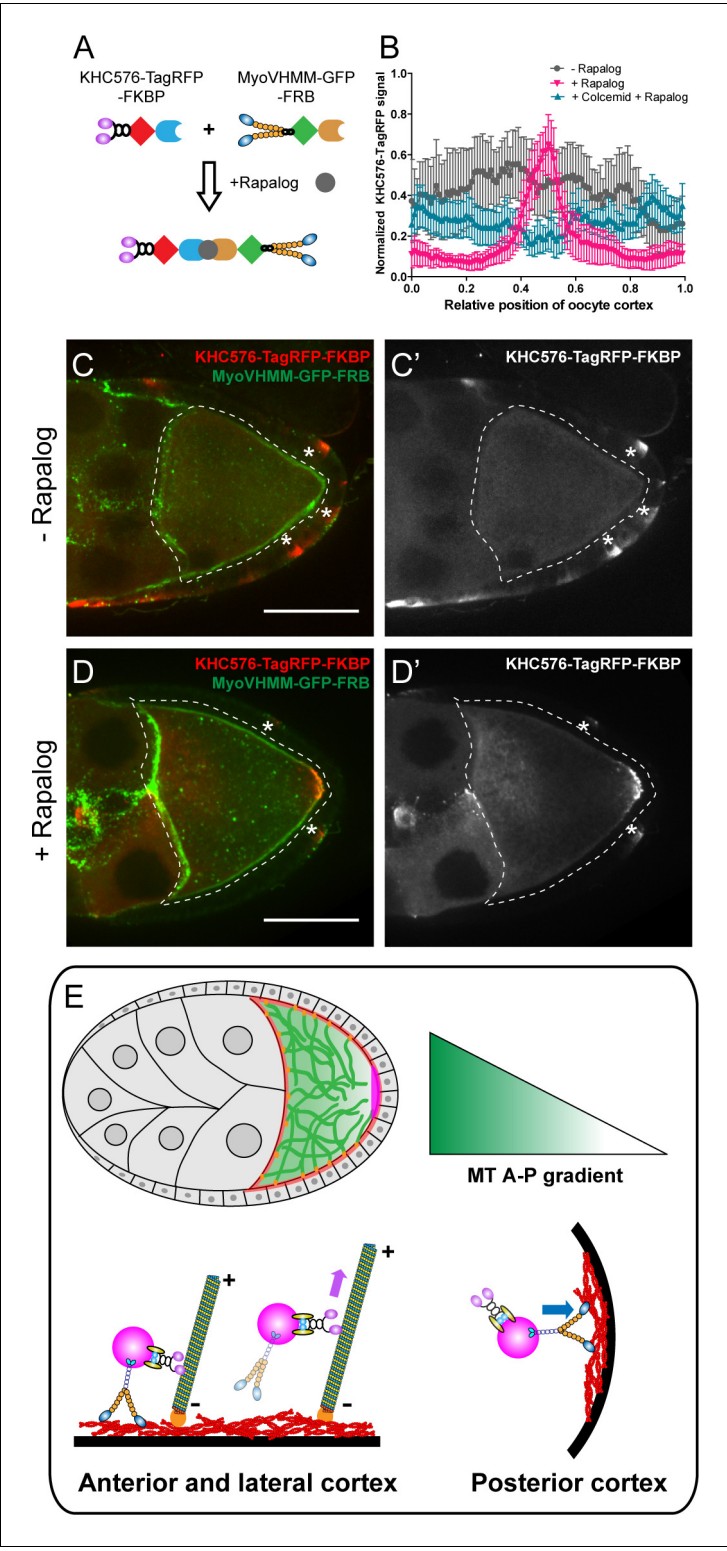

**Figure 5.** Direct competition between kinesin-1 and myosin-V results in the posterior localization. (**A**) A schematic illustration of the rapalog-induced dimerization. Truncated kinesin-1 motor (KHC576, no cargo binding domains) and truncated myosin-V motor (MyoVHMM, no cargo binding domains) are recruited together by rapalog through the FKBP and FRB domains, respectively. (**B**) Normalized KHC576-TagRFP-FKBP signal (average ±95% confidence intervals) along the oocyte cortex (as shown in *Figure 1—figure supplement 1C*; see more details in Materials and methods 'Quantification of fluorescence intensity in the oocytes'). '- Rapalog' (+EtOH control, 1 hr incubation),

*Figure 5 continued on next page*

*Figure 5 continued*

N = 12; '+ Rapalog' (10 μM a/c heterodimer, 1 hr incubation), N = 18; '+Colcemid +Rapalog' (200 μM Colcemid fed for 13–14 hr; 10 μM a/c heterodimer, 1 hr incubation), N = 25. (C–D') Representative images of KHC576-TagRFP-FKBP and MyoVHMM-GFP-FRB in stage 9 oocytes without rapalog treatment (C–C'), and with rapalog treatment (D–D'). To note: (1) As MyoVHMM-GFP-FRB was expressed at a higher level than KHC576-TagRFP-FKBP, residual MyoVHMM-GFP-FRB was observed throughout the cortex after rapalog treatment; (2) The *vasa-Gal4* (II) line has some leakage in the follicle cells (marked as '*'). Within these follicle cells, KHC576-TagRFP was concentrated at the basal level (C'–D'), consistent with the known microtubule polarity in follicle cells (minus-ends binding protein Patronin concentrated at the apical side, while plus-ends EB1 comets grow towards to the basal side *Nashchekin et al., 2016*). Scale bars, 50 μm. (E) At mid-oogenesis (stage 8–9), posterior accumulation of *osk*/Staufen is mainly determined by kinesin-myosin competition. High cortical microtubule density at the anterior and lateral cortex allows kinesin-1 to win over myosin-V and exclude *osk*/Staufen particles away from the cortex, resulting in bulk movement of *osk*/Staufen particles towards the posterior area due to the slight bias of plus-ends direction (shown as the microtubules slightly tilted towards the posterior pole). Meanwhile, lower density of cortical microtubules at the posterior pole favors myosin-mediated cortical anchorage over kinesin-driven cortical exclusion, resulting in the cortical accumulation of *osk*/Staufen particles.

---

KHC576(1–576 aa), TagRFP, FKBP were amplified by PCR and inserted into pMT-A by EcoRI(5')/NotI (3'), NotI(5')/XhoI(3'), and XhoI(5')/XbaI(3'), respectively. KHC576-TagRFP-FKBP was then subcloned into pUASp by KpnI(5')/XbaI(3'). pUASp-MyoVHMM-GFP-FRB: MyoVHMM(1–1100 aa), GFP and FRB were amplified by PCR and inserted into pMT-A by SpeI(5')/NotI(3'), NotI(5')/XhoI(3') and XhoI(5')/XbaI(3'), respectively. MyoVHMM-GFP-FRB was then subcloned into pUASp by SpeI(5')/XbaI(3').

All these pUASp plasmids were sent to BestGene for transposase-mediated P-element insertions, except for the MyoV(Δ1017-1114aa)-FLAG for the PhiC31-mediated integration at the attP64 site (3R, 89B9).

### *Drosophila* genetics

Fly stocks and crosses were kept on standard cornmeal food supplemented with dry active yeast at room temperature (~24–25°C), except for the optogenetic flies were raised and kept at 18 ~ 19°C. The following fly stocks were used in this study: *Khc^ΔHinge2* (II, Dr. Jill C. Wildonger, University of Wisconsin-Madison *Kelliher et al., 2018*); *nos-Gal4-VP16* (III *Van Doren et al., 1998*; *Lu et al., 2012*); *mat αtubGal4-VP16[V2H]* (II, Bloomington *Drosophila* Stock Center #7062); *mat αtubGal4-VP16[V37]* (III, Bloomington *Drosophila* Stock Center #7063); *hs-FLP^[12]* (X, Bloomington *Drosophila* Stock Center #1929 *Chou and Perrimon, 1996*); *FRTG13* (II, Bloomington *Drosophila* stock center # 1956); *FRTG13 ubi-GFP.nls* (II, Bloomington *Drosophila* Stock Center # 5826); *FRT42B didum^88* (II), *UASp-MyoV.FL-GFP* (III) (from Dr. Anne Ephrussi, EMBL *Krauss et al., 2009*); *vasa-Gal4* (II, from Dr. Yukiko Yamashita, University of Michigan); *vasa-Gal4* (III, from Dr. Allan Spradling, Carnegie Institution for Science *DeLuca and Spradling, 2018*), *mat αtub-RFP-Staufen* (III, from Dr. St Johnson, *Parton et al., 2011*); *UASp-GFP-Patronin* (II; from Dr. Uri Abdu, Ben Gurion University); Jupiter-GFP (ZCL2183) (*Morin et al., 2001*); UASp-tdEOS2-αtub84B (II *Lu et al., 2013*). The following fly stocks were generated in this study: *UASp-MyoV (Δ1017–1114 aa)-FLAG* (attP64, III), *UASp-Klp10A-GFP-SspB* (II), *UASp-HA-LIfeAct-SsrA* (II), *UASp-EMTB-3XTagRFP (III)*, *UASp-KHC576-TagRFP-FKBP* (III), and *UASp-MyoVHMM-GFP-FRB* (II). A combined double germline-specific Gal4 driver line (*mat αtub-Gal4-VP16[V2H]; nos-Gal4-VP16*) was used to drive both the RNAi and overexpression, while *mat αtubGal4-VP16[V2H]/+; nosGal4-VP16/+* was used as control.

### Induction of germline clones of *Khc^ΔHinge2*

A standard recombination protocol was performed between *FRTG13* and *Khc^ΔHinge2*. These *FRTG13 Khc^ΔHinge2/CyO* virgin female flies were crossed with males carrying *hs-flp^[12]/y; FRTG13 ubi-GFP.nls/CyO*. From these crosses, young pupae at day 7 and day 8 AEL (after egg laying) were subjected to heat shock at 37°C for 2 hr. Non CyO F1 females were collected 3–4 day after heat shock and fattened with dry active yeast overnight before dissection for Staufen staining.

## Expression of MyoV (Δ1017–1114 aa) in *didum*[88] germline clones

*vasa-Gal4* (III) and *FRTG13 ubi-GFP.nls* were combined to generate *yw; FRTG13 ubi-GFP.nls; vasa-Gal4*. *UASp-MyoV* (Δ1017–1114 aa) (attP64) and *FRT42B didum*[88] were combined to generate *yw; FRT42B didum*[88]*/CyO; UASp-MyoV (Δ1017–1114 aa)* (attP64). *yw,hs-Flp*[12]; *sna*[Sco]*/CyO* virgin female flies were crossed with males of *yw; FRT42B didum*[88]*/CyO; UASp-MyoV (Δ1017–1114 aa)* (attP64) to generate males of *yw,hs-Flp[12]; FRT42B didum*[88]*/CyO; UASp-MyoV (Δ1017–1114 aa)/+*. The males were then crossed with virgin females of *yw; FRTG13 ubi-GFP.nls; vasa-Gal4*. 1st~2nd instar larvae from the cross at day 5 and day 6 AEL (after egg laying) were subjected to heat shock at 37°C water bath for 2 hr. Non CyO non-yellow body color (*UASp-MyoV Δ1017–1114 aa* inserted attP64 carries a *y+* marker) F1 adult females were collected after heat shock and fattened with dry active yeast overnight before dissection for Staufen staining.

## Staufen immunostaining of *Drosophila* oocytes

A standard fixation and staining protocol was used (*Lu et al., 2016*; *Lu et al., 2012*). Samples were incubated with mouse anti-Staufen antibody (1:50, a gift from Dr. Chris Q. Doe, University of Oregon) at 4°C overnight, washed with PBTB (1XPBS + 0.1% Triton X-100 + 0.2% BSA) five times for 10 min each time, incubated with TRITC-conjugated anti-mouse secondary antibody (Jackson ImmunoResearch Laboratories, Inc) at 1:100 at room temperature (24 ~ 25°C) for 4 hr, and washed with PBTB five times for 10 min each before mounting. Samples were imaged either on a Nikon A1plus scanning confocal microscopy with a GaAsP detector and a 40 × 1.30 N.A. oil lens using Galvano scanning, or on a Nikon Eclipse U2000 inverted stand with a Yokogawa CSU10 spinning-disk confocal head and a 40 × 1.30 NA lens using an Evolve EMCCD, both controlled by Nikon Elements software. Images were acquired every 0.5 μm/step in z stacks.

## Quantification of fluorescence intensity in the oocytes

A 5 μm maximum intensity z projection was generated in each sample. The plot profile was either generated along a 3.6 μm-wide line delineating the oocyte cortex (starting and ending at the area where the oocyte meets the nurse cells, *Figure 1—figure supplement 1C*), a 2.5 μm-wide line delineating the oocyte cortex (starting and ending in the middle of the boundary between nurse cells and the oocyte, *Figure 2N*), or an 8 μm-wide line along the anterior–posterior axis (starting the boundary between nurse cells and oocytes, and ending at the posterior-most follicle cells, *Figure 1D*). We normalized the distance in the plot profiles using a custom MatLab program (*Lu et al., 2018*). We also normalized fluorescence intensity (maximum and minimum) in *Figures 1*, *2*, *3* and *5*; *Figure 1—figure supplement 1*; *Figure 2—figure supplement 1*; but not in *Figure 4E*; *Lu, 2020a* and *Lu, 2020b*). Both MatLab codes are now available on GitHub (copies archived at https://github.com/elifesciences-publications/Wen-Lu); normalize both fluorescence intensity and distance (*Lu, 2020a*); normalize distance only (*Lu, 2020b*).

## Recombinant protein production and purification

cDNAs encoding for myosin-V wild-type heavy chain and myosin-V heavy chain deletion (Δ1017–1114 aa) were inserted into a modified pFastBac1 vector which contains a FLAG-tag sequence to the C-terminus. Transposition and the generation of recombinant baculovirus were performed following manufacturer's protocols (Thermo Fisher Scientific). For protein production, Sf9 insect cells were co-infected with recombinant baculovirus encoding for myosin-V heavy chains, *Drosophila* calmodulin and *Drosophila* cytoplasmic myosin light chain (Mlc-c). The protein complex was purified via FLAG affinity chromatography (Sigma) as described (*Billington et al., 2013*; *Melli et al., 2018*) with minor modifications. Eluted proteins were dialyzed overnight against high salt buffer containing 10 mM MOPS, pH 7.2, 500 mM NaCl, 0.1 mM EGTA, 2 mM MgCl$_2$ and 1 mM DTT. Myosins were further concentrated by low speed centrifugation (4000 X g, 15 min) with Amicon filter units (Millipore Sigma) and flash frozen with liquid nitrogen for future use.

## Actin gliding assays

Flow chambers were constructed using a 1% w/v nitrocellulose coated #1.5 coverslip. Buffers were based on a motility buffer (MB) consisting of 20 mM MOPS (pH 7.4), 5 mM MgCl$_2$, 0.1 mM EGTA. Myosin was introduced in high salt buffer (0.3 mg/ml myosin in MB + 500 mM NaCl, 1 mM DTT - 1

min incubation). The surface was blocked with BSA (1 mg/ml in MB + 500 mM NaCl, 1 mM DTT - 1 min). Inactive myosin heads were blocked with unlabeled F-actin (1 µM rabbit skeletal muscle F-actin in MB + 50 mM NaCl, 1 mM ATP - 1 min). The chamber was washed using MB + 50 mM NaCl, 1 mM ATP followed by MB + 50 mM NaCl. 10 nM rhodamine phalloidin-labeled actin was added and allowed to land for 1 min before starting adding the final assay buffer (MB + 50 mM NaCl, 1 mM ATP, 50 mM DTT, 2.5 mg/ml glucose, 100 µg/ml glucose oxidase, 40 µg/ml catalase, 0.1% Methyl-cellulose). Movies were acquired using a Nikon Eclipse Ti-E microscope (temperature maintained at 25°C) at 10 frames per second. Movies were subsequently downsampled 10 fold, allowing for suffi-cient movement between frames to avoid tracking errors. Motility was quantified using the FAST program (*Aksel et al., 2015*). A tolerance filter of 33% was used to exclude intermittently moving fil-aments and a minimum velocity filter of 20 nm/s was used to exclude stuck filaments. For each myo-sin, data from three separate chambers were collected and 3 fields of view were analyzed from each chamber.

## Steady-state ATPase assay

Steady-state ATPase activities were measured in Cary 60 spectrophotometer (Agilent Technologies) at 25°C in buffers containing 10 mM MOPS, pH 7.2, 1 mM ATP, 50 mM NaCl, 2 mM $MaCl_2$, and either 0.1 mM EGTA or 0.1 mM free $CaCl_2$. The buffers also contained 1 µM exogenous calmodulin and an NADH-coupled, ATP-regenerating system including 40 unites/ml lactate dehydrogenase, 200 units/ml pyruvate kinase, 200 µM NADH and 1 mM phosphoenolpyruvate. The rate of ATP hydrolysis was measured from the decrease in absorbance at 340 nm caused by the oxidation of NADH.

## Electron microscopy

Myosins were diluted to a concentration of 50 nM in 10 mM MOPS (pH 7.0), 50 mM NaCl (or 500 mM to visualize the extended conformation), 2 mM $MgCl_2$, 0.1 mM EGTA and 0.1 mM ATP where indicated. To stabilize the inhibited conformation prior to making grids, samples were crosslinked using 0.1% glutaraldehyde for 1 min at room temperature and the reaction was quenched by adding Tris-HCl to a final concentration of 100 mM. Samples were applied to carbon-coated copper grids and stained with 1% uranyl acetate. Micrographs were recorded on a JEOL 1200EXII microscope using an AMT XR-60 CCD camera at a nominal magnification of 60000x. Image processing was per-formed using SPIDER software as described previously (*Burgess et al., 2004*). An initial dataset of 2255 particles (WT myosin V) was aligned and classified into 100 classes using K-means classification. Classes (44/100) which most clearly demonstrated the inhibited conformation were selected and those particles (n = 962) were realigned and classified into 40 classes. Angles measurements were carried out by manual selection of head pairs and corresponding lever-lever junctions in raw images. The selection coordinates were then use to determine the head-junction-head angles for each mole-cule (n = 176 molecules for each myosin). Molecules in which the two heads were superimposed due to being in a side orientation were not selected.

## Immunostaining of microtubules in *Drosophila* oocytes

Ovaries were dissected and gently teased apart in Brinkley Renaturing Buffer 80 (BRB80), pH 6.8 (80 mM piperazine-N,N'-bis(2-ethanesulfonic acid) [PIPES]), 1 mM $MgCl_2$, 1 mM EGTA); permeabilized and extracted in BRB80+1% Triton X-100 at 25°C for 1 hr (without agitation); fixed in 1XPEM (100 mM PIPES pH 6.9, 2 mM EGTA and 1 mM $MgCl_2$) +0.1%Triton X-100+0.5% Glutaraldehyde for 20 min on the rotator; briefly washed with 1XPBS three times; quenched in $NaBH_4$+1XPBS on rotator for 20 min; washed with 1XPBS briefly; washed with 1XPBTB (1XPBS+0.1% Triton X-100+0.2%BSA) five times; blocked with 5% NGS (normal goat serum) in PBTB for 1 hr; stained with the 647-conju-gated anti-tubulin nanobody (a gift from Dr. Helge Ewers, Freie Universität Berlin *Mikhaylova et al., 2015*) 1:50 at 4C overnight; washed with PBTB five times before mounting. Images of nanobody tubulin staining were either imaged on Nikon A1plus scanning confocal microscopy with a GaAsP detector, and a 40 × 1.30 N.A. oil lens using Galvano scanning (*Figure 3*), or Nikon Eclipse U2000 inverted stand with a Yokogawa CSU10 spinning-disk confocal head and a 40 × 1.30 NA lens using an Evolve EMCCD (*Figure 3—figure supplement 1*), both controlled by Nikon Elements software. A single focal plane image was used to present the microtubule distribution.

## Quantification of microtubule staining in oocytes

All the nanobody images were identically imaged and processed. Microtubules (only in the oocyte area, not including nurse cells or follicle cells) were recognized using an ImageJ plugins (Curvetracing, Steger's algoristhm, developed in Cell Biology group of Utrecht University by Jalmar Teeuw and Eugene Katrukha) and the length of all traced lines was measured. The total length of all traced microtubules lines divided by oocyte area was used as an indicator of the microtubule density in each sample.

## Labeling of microtubules by Jupiter-GFP in *Drosophila* oocytes

Ovaries from the stock of Jupiter-GFP (ZCl2183) were dissected and gently teased apart in BRB80 Buffer, pH 6.8; permeabilized and extracted in BRB80+1% Triton X-100 at 25°C for 1 hr (without agitation); fixed in 1XPEM (100 mM PIPES pH 6.9, 2 mM EGTA and 1 mM $MgCl_2$) +0.1%Triton X-100+0.5% Glutaraldehyde for 20 min; briefly washed with 1XPBS three times; quenched in $NaBH_4$+1XPBS on rotator for 20 min; washed with 1XPBS briefly; washed with 1XPBTB (1XPBS+0.1% TritonX-100+0.2% BSA) five times before mounting. Samples were imaged using Nikon A1plus scanning confocal microscopy with a GaAsP detector, and a 40 × 1.30 N.A. oil lens using Galvano scanning, controlled by Nikon Elements software. Images were acquired every 0.5 µm/step in z stacks and a 2.5 µm Maximum intensity projection was used to present the microtubule distribution.

## Labeling of microtubules by tdEOS-αtub in *Drosophila* oocytes

Ovaries from flies expression tdEOS-αtub under maternal αtub-Gal4-VP16 [V37] (mat αtub > tdEOS-αtub) were dissected and gently teased apart in BRB80 Buffer, pH 6.8); permeabilized and extracted in BRB80+1% Triton X-100 at 25°C for 1 hr (without agitation); fixed in 1XPBS +0.1%Triton X-100 +4% EM-grade formaldehyde for 20 min on the rotator; briefly washed with 1XPBS five times before mounting. Samples were imaged using Nikon A1plus scanning confocal microscopy with a GaAsP detector, and a 40 × 1.30 N.A. oil lens using Galvano scanning, controlled by Nikon Elements software. Images were acquired every 0.5 µm/step in z stacks and a 2.5 µm Maximum intensity projection was used to present the microtubule distribution.

## Colcemid treatment

Young newly-eclosed females flies were first fatten with dry active yeast for 18–24 hr (~10–15 females + 5 males in a vial), and then starved for 8 hr. A yeast paste scrambled with 200 µM colcemid or DMSO (control) was then provided to the flies. After 13–14 hr of feeding, flies were dissected for examination.

## Optogenetic recruitment

Flies of following were used in the optogenetic recruiting experiments: (1) *yw; UASp-LifeAct-HA-SsrA/UASp-Klp10A-GFP-SspB; nos-Gal4-VP16/UASp-EMTB-3XTagRFP*; (2) *yw; UASp-LifeAct-HA-SsrA/+; nos-Gal4-VP16/UASp-EMTB-3XTagRFP*; (3) *yw; UASp-LifeAct-HA-SsrA/UASp-Klp10A-GFP-SspB; nos-Gal4-VP16/mat αtub-RFP-Staufen*; (4) *yw; UASp-LifeAct-HA-SsrA /+; nos-Gal4-VP16/mat αtub-RFP-Staufen*. Flies were crossed and raised at 18°C and protected from light to ensure normal oogenesis. Flies were fattened with dry active yeast for 48 hr at 18°C and then dissected in Halocarbon oil 700 (Sigma-Aldrich) as previously described (*Lu et al., 2016*; *Lu et al., 2018*). Dissected samples were protected from light and kept in complete darkness for at least 20 min before imaging. Samples were imaged using Nikon A1plus scanning confocal microscopy with a GaAsP detector, and a 40 × 1.30 N.A. oil lens using Galvano scanning, controlled by Nikon Elements software. For local recruitment, a 15 µm x 15 µm ROI was stimulated by 488 nm laser at 0.06% power (488 nm laser power: 15 mW) for 15 scans at 1 frame per second speed per scan for each recruitment step. For RFP-Staufen imaging, after each recruitment, 561 nm laser were used to image for three frames (2.12 s intervals); the stimulation and imaging procedures were repeated 30 times for total imaging time of 637 s in each samples. For EMTB-3XTagRFP imaging, the recruitment and waiting cycles were identically to the RFP-Staufen imaging; 561 nm laser were used to image once after two recruitment cycles to reduce photobleaching. Fluorescence intensity in the stimulated region of RFP-Staufen and EMTB-3XTagRFP were measured by 'Time measurement' in Nikon Elements, and ImageJ ROI measurement, respectively.

## Rapalog recruitment in oocytes

Young newly-eclosed female flies (*yw; UASp-MyoVHMM-GFP-FRB/vasa-Gal4; UASp-KHC576-TagRFP-FKBP/+*) were fatten with dry active yeast for 18–24 hr (~10–15 females + 5 males in a vial) and then dissected in Xpress insect medium (Lonza). The dissected ovaries were then carefully teased apart using G271/2 syringe needles in a glass bottom dish to loosen up the ovarioles. 10 µM a/c heterodimerizer (Clontech, cat#635057) was added to the medium (the same volume of EtOH was added to the control group). After 1 hr of a/c heterodimerizer incubation, the ovarioles were fixed with 4% EM-grade methanol-free formaldehyde diluted in PBT (1 × PBS and 0.1% Triton X-100) for 20 min on a nutator, and washed five times with PBTB for 10 min each before mounting. Samples were imaged on Nikon Eclipse U2000 inverted stand with a Yokogawa CSU10 spinning-disk confocal head and a 40 × 1.30 NA lens using an Evolve EMCCD controlled by Nikon Elements software. Images were acquired every 0.5 µm/step in Z stacks.

## Acknowledgements

We thank Dr. Chris Q Doe (University of Oregon) for anti-Staufen antibody, Dr. Helge Ewers (Freie Universität Berlin) for tubulin nanobody, Dr. Jill C Wildonger (University of Wisconsin-Madison) for *Khc$^{\Delta hinge2}$* line, Dr. Anne Ephrussi (EMBL) for *FRT42B didum$^{88}$*, *UASp-MyoV.FL-GFP* lines, Dr. Yukiko Yamashita (University of Michigan) for *vasa-Gal4* (II) line, Dr. Allan C Spradling (Carnegie Institution for Science) for *vasa-Gal4* (III) line, Dr. Uri Abdu (Ben-Gurion University) for *UASp-GFP-Patronin* line, Dr. Daniel St Johnston (University of Cambridge) for *mat αtub-RFP-Staufen* line, the Bloomington *Drosophila* Stock Center (supported by National Institutes of Health grant P40OD018537) for fly stocks, and *Drosophila* Genomics Resource Center (supported by NIH grant 2P40OD010949) for cDNA clones. We thank the NHLBI Electron Microscopy Core Facility for the use of their microscopes. We also thank the Gelfand laboratory members for support, discussion, and suggestions. Research reported in this study was supported by the National Institute of General Medical Sciences grants R01GM124029 and R35GM131752 to VI Gelfand.

## Additional information

### Funding

| Funder | Grant reference number | Author |
| --- | --- | --- |
| National Institute of General Medical Sciences | GM124029 | Vladimir I Gelfand |
| National Institute of General Medical Sciences | GM131752 | Vladimir I Gelfand |

The funders had no role in study design, data collection and interpretation, or the decision to submit the work for publication.

### Author contributions

Wen Lu, Conceptualization, Data curation, Formal analysis, Validation, Visualization, Methodology, Writing - original draft, Writing - review and editing; Margot Lakonishok, Rong Liu, Neil Billington, Data curation, Formal analysis, Validation, Visualization, Methodology; Ashley Rich, Michael Glotzer, Resources, Methodology, Writing - review and editing; James R Sellers, Resources, Methodology; Vladimir I Gelfand, Conceptualization, Resources, Supervision, Funding acquisition, Investigation, Writing - original draft, Project administration, Writing - review and editing

### Author ORCIDs

Wen Lu (iD) https://orcid.org/0000-0002-8849-8100
Neil Billington (iD) http://orcid.org/0000-0003-2306-0228
Michael Glotzer (iD) http://orcid.org/0000-0002-8723-7232
James R Sellers (iD) http://orcid.org/0000-0001-6296-564X
Vladimir I Gelfand (iD) https://orcid.org/0000-0002-6361-2798

Decision letter and Author response
Decision letter https://doi.org/10.7554/eLife.54216.sa1
Author response https://doi.org/10.7554/eLife.54216.sa2

## Additional files

### Supplementary files
• Transparent reporting form

### Data availability
All data generated or analysed during this study are included in the manuscript and supporting files.

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
