## [Decision Letter]

**Acceptance summary:**

mRNA localization is a critical mechanism to determine the oocyte polarity (future body plan and germ line determination) during *Drosophila* oogenesis. Although genetic studies have indicated antagonistic roles of Kinesin-1 and myosin-V in *oskar* and staufen localization, there remained puzzles as to how this can be achieved, such as why myosin-V that shows even distribution can specifically localize *oskar*/staufen at a specific (posterior) site in the oocyte. The manuscript by Lu et al. uses elegant experimental approaches to solve these puzzles, bringing about a unified model. The authors responded well to the points raised by reviewers, who all agreed that the manuscript represents a significant contribution to the field.

**Decision letter after peer review:**

Thank you for submitting your article "Competition between kinesin-1 and myosin-V defines *Drosophila* posterior determination" for consideration by *eLife*. Your article has been reviewed by three peer reviewers, including Yukiko M Yamashita as the Reviewing Editor and Reviewer #1, and the evaluation has been overseen by Anna Akhmanova as the Senior Editor. The following individual involved in review of your submission has agreed to reveal their identity: Paul Lasko (Reviewer #3).

The reviewers have discussed the reviews with one another and the Reviewing Editor has drafted this decision to help you prepare a revised submission.

Summary

The reviewers are overall positive, and provided suggestions to improve the clarity of the paper. Except for point 1 by reviewer 2, all comments can be addressed by modifications to texts. Thus, we would like to invite you to submit a revised version of the manuscript. In doing so, we would like you to pay special attention to describing/summarizing past studies. In some cases, the description of past work was not sufficient, failing to convey what is exactly novel/unique about this work.

Reviewer #1:

mRNA localization is a critical mechanism to determine the oocyte polarity (future body plan and germ line determination) during *Drosophila* oogenesis. Although genetic studies have indicated antagonistic roles of Kinesin-1 and myosin-V in *oskar* and staufen localization, there remained puzzles as to how this can be achieved, such as why myosin-V that shows even distribution can specifically localize *oskar*/staufen at a specific (posterior) place in the oocyte. In this study, Gelfand and colleagues show that the competition between microtubule-mediated transport by kinesin-1 and actin-based anchoring by Myosin-V determines posterior localization of *oskar* and staufen. This is primarily determined by density of microtubules at the cortex.

Their conclusion is well-supported by well-executed experiments. In a sense, this paper's conclusion can be predicted by synthesizing previous studies but this is the first to really nail the mechanisms. And the feeling of 'it could have been predicted' only reflects the elegance of this work (often, a very clear study gives this impression-'we should have known!') instead of the lack of novelty. Optogenetic manipulation of microtubule-density at a local spot of oocyte is particularly elegant. I really do not have any major comments, and I think this is an important contribution to the field.

Reviewer #2:

The manuscript by Lu et al. uses elegant experimental approaches to tie up loose ends about how *oskar* RNA becomes localized to the posterior of the *Drosophila* oocyte in order to create the germ plasm. This in turn is essential for the patterning of the embryonic A-P body axis and the development of the germline. To date, there is good evidence that *oskar* is transported by kinesin motors on a weakly polarized microtubule cytoskeleton, resulting in a biased random walk that leads to its ultimate accumulation at the posterior pole. There is also evidence that the retention (anchoring) of *oskar* RNA at the posterior of the oocyte depends on the cortical actin cytoskeleton and the MyoV motor. This work answers the question of why *oskar* becomes anchored only at the posterior when actin and MyoV are uniformly distributed around the periphery of the oocyte. The idea that kinesin drives *oskar* away from the cortex, where it can otherwise "stick" has already been proposed so the paper does not provide a major conceptual leap. But it does uncover the molecular underpinnings that support this model through the use of constructs that allow the authors to manipulate kinesin and MyoV activity as well as the use of optogenetic recruitment of a depolymerizing kinesin to locally deplete microtubules at the cortex and the use of a dimerized Khc-MyoV to test their competition model. The latter two experiments, and particularly the local depletion of microtubules, are the most compelling and provide the most conclusive evidence for a tug of war between kinesin and MyoV.

Overall the data are solid, although a number of issues need to be addressed. Most of these relate to interpretations of results and/or conclusions drawn (see below). In addition, there are many instances where the authors need to be more careful about citing or interpreting previous work.

1) In Figure 1—figure supplement 1B, the cortical distribution of Staufen is not very obvious – this same pattern of fluorescence can be generated with uniformly distributed protein due to edge effects. Cortical localization of *oskar* and Staufen in khc germline clones has been more convincingly documented by others. Why don't the authors perform FISH for *oskar*, which would provide superior resolution to anti-Staufen IF.

Similarly, in Figures 2L-N and particularly in Figure 3H, it isn't at all clear whether Staufen is really cortical or uniformly distributed throughout the oocyte. The authors should confirm using *oskar* FISH.

2) The distribution shown in Figure 1—figure supplement 1E, F could be equally well explained by delayed localization. The "bump" in the middle is consistent with a delay as is the finding that the distribution at stage 10 is wild-type. Also, the term displaced indicates that *oskar* was localized and then removed from the posterior, which is most likely not the case.

Further consistent with a delay in localization is the observation that "later in development (at stage 10B) normal Staufen distribution is recovered with the restoration of the posterior cap and clearing of the central cytoplasmic aggregate (Figure 1F-1H; Figure 1—figure supplement 1G-1H)".

Similarly, the conclusion: "…we show that constitutively active kinesin-1 causes *osk*/Staufen detachment from the posterior pole (Figure 1),…" is not correct." The data show that Staufen is not properly localized at the posterior, not that it has become detached.

3) The authors focus a lot of attention on the idea of a microtubule gradient, but they don't actually test whether a graded distribution of microtubules is relevant – they only test the differential between having microtubules and not having microtubules.

See also: "Together, these data demonstrate that *osk*/Staufen localization is controlled by the microtubule gradient in the oocyte. Decreased density of cortical microtubules favors myosin-V resulting in uniform cortical localization of *osk*/Staufen, whereas increased microtubule density favors kinesin-1 resulting in exclusion of *osk*/Staufen from the cortex."

If the gradient was relevant, one would expect to see a graded distribution of Staufen at the cortex of wild-type oocytes. The wording should be changed to reflect whether there are/aren't microtubules rather the gradient of microtubules.

4) Figure 5D: It looks like much more than "some cortical localization of MyoVHMM remains at the anterior and lateral cortex. Shouldn't it accumulate around the entire cortex (like wild-type MyoV) if it is not dimerized with KHC576? Why is there such a high concentration specifically at the anterior?

5) Introduction section: The authors state that posterior localization of *oskar* "is maintained throughout late oogenesis (stage 10B to stage 13)". Oogenesis progresses through 14 stages and *oskar* is maintained through stage 14.

6) The statement that Staufen is "commonly used as a marker for posterior determination" is not correct. This should be changed to "Staufen is commonly used as a proxy for *oskar* RNA".

7) The statement that "kinesin-1 has been proposed to drive the *osk* mRNA transport along these cortical microtubules from the anterior to the posterior pole" is an oversimplified and misleading interpretation of published data. In addition, it could easily be taken to mean that microtubules span the length of the oocyte from anterior to posterior pole. The current model indicates that *oskar* is localized by a random walk on a weakly polarized microtubule cytoskeleton. Zimyanin et al., 2008, showed that *oskar* RNA "is actively transported along microtubules in all directions, with a slight bias toward the posterior". In addition, Ghosh et al., 2012, analyzed the motility of *oskar* RNA and derivatives with mutations in localization elements and concluded that for both the wild-type and mutants: "we did not detect a statistically significant net posterior vector."

Same issue with the sentence in the Discussion.

8) The statement that "Kinesin-driven microtubule sliding generates the force that drives fast ooplasmic streaming of the ooplasm in late-stage *Drosophila* oocytes" is an oversimplification of their own lab's work. In Lu et al., 2016, the authors conclude that "This microtubule motility and kinesin-1-based organelle transport together generate the forces required to drive oocyte cytoplasmic streaming".

9) The statement "Streaming, rather than directed transport along microtubules, is responsible for the localization of posterior determinants during late oogenesis" should be referenced to Glotzer et al., 1997 and Forrest and Gavis, 2003. In addition, it is diffusion facilitated by streaming, not streaming per se that is responsible.

10) For the statement "An actin motor, myosin-V (didum in *Drosophila*), is involved in *osk* mRNA cortical localization," the citation should include Sinsimer et al., 2013, in addition to Krauss et al.

11) The statement that "This finding agrees with a previous study using ectopic expression of Khc^ΔHinge2^ in the Khc null background" is incorrect. The cited study study concluded that there was a very minor *oskar* localization defect and that "Our data suggest that KHC auto-inhibition does not play a major role in the oocyte."

12) The wording of the clause "while the decrease of microtubule density at the lateral cortex leads to ectopic accumulation of Staufen at the cortex (Figure 4)" is confusing. The correct grammar is: "whereas reducing microtubule density…."

Reviewer #3:

The paper by Lu et al. constitutes a substantial advance in our understanding of the mechanism of localization of *oskar* RNA and Staufen protein to the posterior pole of the *Drosophila* oocyte. A great deal of previous work supports the conclusion that initial *osk*/Stau localization results from kinesin-driven microtubule-dependent transport, while stabilization of posterior *osk*/Stau is F-actin dependent. This study provides compelling evidence for a unified model to explain this, by demonstrating that *osk*/Stau accumulates in regions where microtubules are present at low concentrations, but that stabilization of *osk*/Stau depends upon Myosin V. The optogenetic experiments in Figure 4 that show that local destabilization of microtubules results in recruitment of Stau to the site of destabilization and the demonstration that a minimal KHC-Myosin V system can support posterior localization represent important advances that significantly extend knowledge in the field.

Nevertheless, I have several concerns that I believe should be addressed prior to publication.

1) Different publications have reached different conclusions about the shallowness or depth of the anterior-posterior microtubule gradient. Even in this paper, there is a major difference between how the gradient looks when it is imaged indirectly using an RFP reporter (Figure 3A) or when endogenous microtubules are directly imaged (Figure 3D). The gradient of the endogenous microtubules appears far less pronounced, as was also reported by Zimyanin et al. (ref 16). Might the reporter accentuate the depth of the gradient? The authors should discuss this issue.

2) More information should be given about the ultimate phenotypic consequences of the genetic manipulations. In particular, do Khc^ΔHinge2^ oocytes (Figure 1G) complete development and support embryogenesis? If so are the resulting embryos normal or posterior-group in phenotype? Inclusion of these data is important for understanding whether subsequent localization mechanisms dependent upon cytoplasmic streaming can fully compensate for abrogation of the mechanism described here.

3) The authors should discuss why they think Stau aggregates in the centre of the oocyte when microtubule density is increased. Also, does the central aggregation of Stau in OE Patronin oocytes (Figure 3I) later resolve itself as it does for Khc^ΔHinge2^? If not can the authors speculate as to why not, and does this affect the argument they propose at the end of the third paragraph in subsection “Staufen localization is controlled by kinesin-1”.

4) In Figure 1—figure supplement 1, control images should be shown for panels E and G, also kinesin is misspelled in the title to this figure.

---

## [Author Response]

The reviewers are overall positive, and provided suggestions to improve the clarity of the paper. Except for point 1 by reviewer 2, all comments can be addressed by modifications to texts. Thus, we would like to invite you to submit a revised version of the manuscript. In doing so, we would like you to pay special attention to describing/summarizing past studies. In some cases, the description of past work was not sufficient, failing to convey what is exactly novel/unique about this work.Reviewer #1:mRNA localization is a critical mechanism to determine the oocyte polarity (future body plan and germ line determination) during *Drosophila* oogenesis. Although genetic studies have indicated antagonistic roles of Kinesin-1 and myosin-V in oskar and staufen localization, there remained puzzles as to how this can be achieved, such as why myosin-V that shows even distribution can specifically localize oskar/staufen at a specific (posterior) place in the oocyte. In this study, Gelfand and colleagues show that the competition between microtubule-mediated transport by kinesin-1 and actin-based anchoring by Myosin-V determines posterior localization of oskar and staufen. This is primarily determined by density of microtubules at the cortex.Their conclusion is well-supported by well-executed experiments. In a sense, this paper's conclusion can be predicted by synthesizing previous studies but this is the first to really nail the mechanisms. And the feeling of 'it could have been predicted' only reflects the elegance of this work (often, a very clear study gives this impression-'we should have known!') instead of the lack of novelty. Optogenetic manipulation of microtubule-density at a local spot of oocyte is particularly elegant. I really do not have any major comments, and I think this is an important contribution to the field.Reviewer #2:The manuscript by Lu et al. uses elegant experimental approaches to tie up loose ends about how oskar RNA becomes localized to the posterior of the *Drosophila* oocyte in order to create the germ plasm. This in turn is essential for the patterning of the embryonic A-P body axis and the development of the germline. To date, there is good evidence that oskar is transported by kinesin motors on a weakly polarized microtubule cytoskeleton, resulting in a biased random walk that leads to its ultimate accumulation at the posterior pole. There is also evidence that the retention (anchoring) of oskar RNA at the posterior of the oocyte depends on the cortical actin cytoskeleton and the MyoV motor. This work answers the question of why oskar becomes anchored only at the posterior when actin and MyoV are uniformly distributed around the periphery of the oocyte. The idea that kinesin drives oskar away from the cortex, where it can otherwise "stick" has already been proposed so the paper does not provide a major conceptual leap. But it does uncover the molecular underpinnings that support this model through the use of of constructs that allow the authors to manipulate kinesin and MyoV activity as well as the use of optogenetic recruitment of a depolymerizing kinesin to locally deplete microtubules at the cortex and the use of a dimerized Khc-MyoV to test their competition model. The latter two experiments, and particularly the local depletion of microtubules, are the most compelling and provide the most conclusive evidence for a tug of war between kinesin and MyoV.Overall the data are solid, although a number of issues need to be addressed. Most of these relate to interpretations of results and/or conclusions drawn (see below). In addition, there are many instances where the authors need to be more careful about citing or interpreting previous work.1) In Figure 1—figure supplement 1B, the cortical distribution of Staufen is not very obvious – this same pattern of fluorescence can be generated with uniformly distributed protein due to edge effects. Cortical localization of oskar and Staufen in khc germline clones has been more convincingly documented by others. Why don't the authors perform FISH for oskar, which would provide superior resolution to anti-Staufen IF.Similarly, in Figures 2L-N and particularly in Figure 3H, it isn't at all clear whether Staufen is really cortical or uniformly distributed throughout the oocyte. The authors should confirm using oskar FISH.

We thank the reviewer for this comment. Here we have to admit that our original cortical Staufen staining in KHC-RNAi mutant is not as obvious as previously published *Khc* germline clones using *osk* FISH (e.g., Cha et al., 2002). Based on our *osk* FISH staining in the previously published study, we observed no major difference in image quality between *osk* FISH and anti-Staufen IF (Lu et al., 2018, compared Figure S4S-T (*osk* FISH) to Figure S5F-G (anti-Staufen IF)). We think the major reason for inferior cortical Staufen staining in the original manuscript was that we used a spinning disk confocal microscopy for the Staufen imaging, which resulted in sub-optimal z-resolution. To improve the image quality, we re-imaged the Staufen staining using A1-plus confocal scanning microscope with a GaAsP detector. We found better examples of the uniform cortical Staufen localization in KHC-RNAi and Klp10A overexpressing samples and partial cortical Staufen staining in MyoVΔ1017-1114 overexpression samples. As a result, we have updated Figure 2, Figure 3 and Figure 1—figure supplement 1 with the new Staufen staining images.

2) The distribution shown in Figure 1—figure supplement 1E, F could be equally well explained by delayed localization. The "bump" in the middle is consistent with a delay as is the finding that the distribution at stage 10 is wild-type. Also, the term displaced indicates that oskar was localized and then removed from the posterior, which is most likely not the case.Further consistent with a delay in localization is the observation that "later in development (at stage 10B) normal Staufen distribution is recovered with the restoration of the posterior cap and clearing of the central cytoplasmic aggregate (Figure 1F-1H; Figure 1—figure supplement 1G-1H)".Similarly, the conclusion: "…we show that constitutively active kinesin-1 causes osk/Staufen detachment from the posterior pole (Figure 1),…" is not correct." The data show that Staufen is not properly localized at the posterior, not that it has become detached.

We thank the reviewer for this comment. We have replaced “displace” with “mislocalize”, and replaced “detachment” with “mislocalization”.

Regarding the possible delayed localization instead of mislocalization, we believe it is less likely for two reasons:

1) At stage 8 (before the posterior localization is achieved), we never observed Staufen staining as a “bump” (cytoplasmic aggregation) in the control; instead it is “cloudy” cytoplasmic localization. Therefore, Staufen aggregation in the cytoplasm in *Khc^Δhinge2^* clone (stage 9) does not resemble the control at the earlier stage (stage 8).

2) The rescue of Staufen localization observed in *Khc^Δhinge2^* 10B oocytes is mostly driven by streaming circulation. At stage 10B, the microtubules reorganize dramatically and it would be very difficult to achieve the posterior accumulation based on kinesin-driven transport at stage 10B.

3) The authors focus a lot of attention on the idea of a microtubule gradient, but they don't actually test whether a graded distribution of microtubules is relevant – they only test the differential between having microtubules and not having microtubules.See also: "Together, these data demonstrate that osk/Staufen localization is controlled by the microtubule gradient in the oocyte. Decreased density of cortical microtubules favors myosin-V resulting in uniform cortical localization of osk/Staufen, whereas increased microtubule density favors kinesin-1 resulting in exclusion of osk/Staufen from the cortex."If the gradient was relevant, one would expect to see a graded distribution of Staufen at the cortex of wild-type oocytes. The wording should be changed to reflect whether there are/aren't microtubules rather the gradient of microtubules.

We agree with the reviewer that what controls Staufen localization is the absolute density of microtubule tracks. Based on our optogenetic experiment, local microtubule density is the key to determine whether Staufen would be cortically localized. To avoid any confusion, we have replaced the “microtubule gradient” with “microtubule density” discussing the motor competition. We still keep the term “microtubule gradient” in other contents (Introduction and reference to other’s work, etc) just to keep it consistent with previously published literature.

4) Figure 5D: It looks like much more than "some cortical localization of MyoVHMM remains at the anterior and lateral cortex. Shouldn't it accumulate around the entire cortex (like wild-type MyoV) if it is not dimerized with KHC576? Why is there such a high concentration specifically at the anterior?

In fact, for the MyoVHMM channel, we did observe accumulation around the entire cortex (like wild-type MyoV). The slightly higher concentration at the anterior cortex could be due to its proximity to the ring canals where the MyoVHMM is transported to the oocyte. Here we used the term “at the anterior and lateral cortex” to distinguish from the posterior cortex where KHC576 accumulates after rapalog treatment.

5) Introduction section: The authors state that posterior localization of oskar "is maintained throughout late oogenesis (stage 10B to stage 13)". Oogenesis progresses through 14 stages and oskar is maintained through stage 14.

The reviewer is correct; we have corrected to stage 14.

6) The statement that Staufen is "commonly used as a marker for posterior determination" is not correct. This should be changed to "Staufen is commonly used as a proxy for oskar RNA".

We have corrected the statement.

7) The statement that "kinesin-1 has been proposed to drive the osk mRNA transport along these cortical microtubules from the anterior to the posterior pole" is an oversimplified and misleading interpretation of published data. In addition, it could easily be taken to mean that microtubules span the length of the oocyte from anterior to posterior pole. The current model indicates that oskar is localized by a random walk on a weakly polarized microtubule cytoskeleton. Zimyanin et al., 2008, showed that oskar RNA "is actively transported along microtubules in all directions, with a slight bias toward the posterior". In addition, Ghosh et al., 2012, analyzed the motility of oskar RNA and derivatives with mutations in localization elements and concluded that for both the wild-type and mutants: "we did not detect a statistically significant net posterior vector."

We have modified the Introduction to better reflect the current model in the field: “The anterior-posterior gradient of cortical microtubules results in slightly more microtubule plus ends oriented towards the posterior pole. Kinesin-1 has been proposed to drive *osk* mRNA transport along these weakly biased cortical microtubules, favoring *osk* mRNA movement from the anterior to the posterior pole”

Same issue with the sentence in the Discussion.

We have also modified the Discussion to clarify the model better: “Previously, several groups have proposed that kinesin-1 transports *osk*/Staufen particles along slightly biased cortical microtubules, resulting in net movement of *osk*/Staufen from the anterior side to the posterior pole in stage 8-9 oocytes”.

8) The statement that "Kinesin-driven microtubule sliding generates the force that drives fast ooplasmic streaming of the ooplasm in late-stage *Drosophila* oocytes" is an oversimplification of their own lab's work. In Lu et al., 2016, the authors conclude that "This microtubule motility and kinesin-1-based organelle transport together generate the forces required to drive oocyte cytoplasmic streaming".

We have corrected the statement in the Introduction to: “Kinesin-driven microtubule sliding together with kinesin-driven cargo transportgenerates the force that drives fast cytoplasmic streaming of the ooplasm in late-stage *Drosophila* oocytes”

9) The statement "Streaming, rather than directed transport along microtubules, is responsible for the localization of posterior determinants during late oogenesis" should be referenced to Glotzer et al., 1997 and Forrest and Gavis, 2003. In addition, it is diffusion facilitated by streaming, not streaming per se that is responsible.

We apologize for missing out these two key references. We have corrected the text and added the two citations according to the reviewer’s suggestion.

10) For the statement "An actin motor, myosin-V (didum in *Drosophila*), is involved in osk mRNA cortical localization," the citation should include Sinsimer et al., 2013, in addition to Krauss et al.

We have added the reference of Sinsimer et al., 2013.

11) The statement that "This finding agrees with a previous study using ectopic expression of Khc^ΔHinge2^ in the Khc null background" is incorrect. The cited study study concluded that there was a very minor oskar localization defect and that "Our data suggest that KHC auto-inhibition does not play a major role in the oocyte."

Although Williams et al., 2014 paper concluded that”KHC auto-inhibition does not play a major role in the oocyte”, it is clear that they did observe some effects of KHC1-975^ΔHinge2^ on Staufen localization (please compare their Figure 1B and Figure 7A). Obviously, our phenotype is stronger than what was observed by William et al. We believe that the main reason for that is that we used the endogenous locus for knock-in hingeless kinesin-1, so the level and timing of our Khc^∆hinge2^ expression match that of the wild-type motor. As a result, we get better penetration of the phenotype.

To reflect this difference, we modified our text as follows: “The endogenous deletion of Khc hinge2 region shows a stronger phenotype of Staufen mislocalization than the ectopic expression of *Khc^ΔHinge2^*in the*Khc* null background from a previous study”.

12) The wording of the clause "while the decrease of microtubule density at the lateral cortex leads to ectopic accumulation of Staufen at the cortex (Figure 4)" is confusing. The correct grammar is: "whereas reducing microtubule density…."

We thank the reviewer for this comment and corrected the sentence.

Reviewer #3:The paper by Lu et al. constitutes a substantial advance in our understanding of the mechanism of localization of oskar RNA and Staufen protein to the posterior pole of the *Drosophila* oocyte. A great deal of previous work supports the conclusion that initial osk/Stau localization results from kinesin-driven microtubule-dependent transport, while stabilization of posterior osk/Stau is F-actin dependent. This study provides compelling evidence for a unified model to explain this, by demonstrating that osk/Stau accumulates in regions where microtubules are present at low concentrations, but that stabilization of osk/Stau depends upon Myosin V. The optogenetic experiments in Figure 4 that show that local destabilization of microtubules results in recruitment of Stau to the site of destabilization and the demonstration that a minimal KHC-Myosin V system can support posterior localization represent important advances that significantly extend knowledge in the field.Nevertheless, I have several concerns that I believe should be addressed prior to publication.1) Different publications have reached different conclusions about the shallowness or depth of the anterior-posterior microtubule gradient. Even in this paper, there is a major difference between how the gradient looks when it is imaged indirectly using an RFP reporter (Figure 3A) or when endogenous microtubules are directly imaged (Figure 3D). The gradient of the endogenous microtubules appears far less pronounced, as was also reported by Zimyanin et al. (ref 16). Might the reporter accentuate the depth of the gradient? The authors should discuss this issue.

We agree with the reviewers that the endogenous tubulin staining shows less pronounced gradient than the minus-end Patronin-GFP and the MAP EMTB-TagRFP. The main reason for this discrepancy is that tubulin immunostaining in addition to microtubules also detects the soluble tubulin pool (which, obviously, does not form the gradient). To overcome this problem, we use a protein trap line of an endogenous MAP (Jupiter-GFP, ZCL2183) to visualize microtubules in the oocyte. Using Jupiter-GFP, we demonstrated a clear anterior-posterior gradient of microtubules at stage 9 (added as Figure 3—figure supplement1A).

2) More information should be given about the ultimate phenotypic consequences of the genetic manipulations. In particular, do Khc^ΔHinge2^ oocytes (Figure 1G) complete development and support embryogenesis? If so are the resulting embryos normal or posterior-group in phenotype? Inclusion of these data is important for understanding whether subsequent localization mechanisms dependent upon cytoplasmic streaming can fully compensate for abrogation of the mechanism described here.

The *Khc^Δhinge2^* mutant oocytes have no obvious defects during late oogenesis (such as pole cell specification) and no overall embryonic lethality. This is consistent with the fact that the posterior localization of Staufen is restored in stage 10B (probably by streaming circulation). Cytoplasmic streaming is fully capable of rescuing *osk*/Stau mislocalization occurred in mid-oogenesis and rescuing pole cell specification and embryonic viability, which we published in a previous study (Lu et al., 2018). We believe that detailed discussion of these data would distract the readers from the main message of this paper.

3) The authors should discuss why they think Stau aggregates in the centre of the oocyte when microtubule density is increased. Also, does the central aggregation of Stau in OE Patronin oocytes (Figure 3I) later resolve itself as it does for Khc^ΔHinge2^? If not can the authors speculate as to why not, and does this affect the argument they propose at the end of the third paragraph in subsection “Staufen localization is controlled by kinesin-1”.

According to our model, *osk*/Staufen aggregation in Patronin overexpression is caused by increased microtubule density at the posterior pole, which in turn allows kinesin-1 to win over myosin-V and translocate the *osk*/Staufen particles from the posterior pole to the microtubules plus-ends. As most of the microtubule plus-ends are pointing towards the centre of the oocyte, *osk*/Staufen accumulates in the middle of the oocyte. We have included a brief explanation of the cytoplasmic aggregation in Patronin overexpression: “This increase of microtubule density causes the exclusion of Staufen from the entire cortex and forms aggregation in the center of the oocyte cytoplasm where most of the plus-ends of microtubules are pointing towards (Figure 3G, 3I and 3K).”.

Different from the *Khc^∆hinge2^* mutant, these Patronin O/E oocytes do not restore the posterior cap. Instead, small Staufen aggregates get scattered in the cytoplasm at stage 10B, probably due to the shear force generated by streaming (see the attached image of Staufen staining at late stage 10B with Patronin overexpression). We speculate that the difference between *Khc^∆hinge2^* mutant and Patronin O/E in late stages lies in the *osk*/Staufen cap formed at stage 9. Compared the Figure 1 (C and E) and Figure 3 (I and K), we found that the *Khc^∆hinge2^* mutant forms a cap that is at a comparable intensity level to the cytoplasmic aggregation, while most of the Staufen is in the cytoplasmic aggregation in Patronin O/E mutant. This mini-cap at stage 9 in *Khc^∆hinge2^* mutant can translate into Osk protein and therefore forms a positive feedback loop to attract more *osk*/Staufen during streaming. We have a brief explanation in the Results: “Previously, studies suggest that Osk protein, translated at the posterior pole, functions in a positive feedback mechanism for *osk*/Staufen accumulation in streaming oocytes. We postulate that enough Osk protein translates from the residual cap, initiating the positive feedback loop, while ooplasmic streaming circulates mislocalized *osk*/Staufen particles to the posterior cap, enhancing Osk localization, resulting the restoration of the posterior crescent.”

In contrast, there may not be enough *osk* mRNA at the posterior pole of Patronin O/E mutant at stage 9 and therefore it fails to initiate the positive feedback loop using translated Osk protein. In this case, the big cytoplasmic aggregation gets broken into small aggregations by streaming, but they are never able to get recruited to the posterior cap.

4) In Figure 1—figure supplement 1, control images should be shown for panels E and G, also kinesin is misspelled in the title to this figure.

We have added the control images in Figure 1—figure supplement 1E-1F.

The typo “Kinesin” in the figure title has been corrected.